# Generalizing Denoising to Non-Equilibrium Structures Improves Equivariant Force Fields

## Abstract

Understanding the interactions of atoms such as forces in 3D atomistic systems is fundamental to many applications like molecular dynamics and catalyst design. However, simulating these interactions requires compute-intensive *ab initio* calculations and thus results in limited data for training neural networks. In this paper, we propose to use **de**noising **n**on-equilibrium **s**tructures (**DeNS**) as an auxiliary task to better leverage training data and improve performance. For training DeNS, we first corrupt a 3D structure by adding noise to its 3D coordinates and then predict the noise. Different from previous works on pre-training via denoising, which are limited to equilibrium structures, the proposed DeNS generalizes to a much larger set of non-equilibrium structures without relying on another dataset for pre-training. The key enabler is the encoding of input forces. A non-equilibrium structure has non-zero forces and thus many possible atomic positions, making denoising an ill-posed problem. To address the issue, we additionally take the forces of the original structure as inputs to specify which non-equilibrium structure we are denoising. Concretely, given a corrupted non-equilibrium structure and the forces of the original one, we predict the non-equilibrium structure satisfying the input forces instead of any arbitrary structures. Since DeNS requires encoding forces, DeNS favors equivariant networks, which can easily incorporate forces and other higher-order tensors in node embeddings.

We demonstrate the effectiveness of training equivariant networks with DeNS on OC20, OC22 and MD17 datasets. For OC20, EquiformerV2 (Liao et al., 2023) trained with DeNS achieves better Structure to Energy and Forces (S2EF) results and comparable Initial Structure to Relaxed Energy (IS2RE) results. For OC22, EquiformerV2 trained with DeNS establishs new state-of-the-art results. For MD17, Equiformer ($L_{max} = 2$) (Liao & Smidt, 2023) trained with DeNS achieves better results and saves $3.1\times$ training time compared to Equiformer ($L_{max} = 3$) without DeNS, where $L_{max}$ denotes the maximum degree. We also show that DeNS can improve other equivariant networks like eSCN (Passaro & Zitnick, 2023) on OC20 and SEGNN-like networks (Brandstetter et al., 2022) on MD17.

## 1 Introduction

Graph neural networks (GNNs) have made remarkable progress in approximating high-fidelity, compute-intensive quantum mechanical calculations like density functional theory (DFT) for atomistic systems (Gilmer et al., 2017; Zhang et al., 2018; Unke et al., 2021; Batzner et al., 2022; Rackers et al., 2023; Lan et al., 2022), enabling new insights in applications such as molecular dynamics simulations (Musaelian et al., 2023) and catalyst design (Chanussot* et al., 2021; Lan et al., 2022). However, unlike other domains such as natural language processing (NLP) and computer vision (CV), the scale of atomistic data is quite limited since generating data requires compute-intensive *ab initio* calculations. For example, the largest atomistic dataset, OC20 (Chanussot* et al., 2021), contains about 138M examples while GPT-3 (Brown et al., 2020) is trained on hundreds of billions of words and ViT-22B (Dehghani et al., 2023) is trained on around 4B images.

To start addressing this gap, we take inspiration from self-supervised learning methods in NLP and CV and explore how we can adapt them to learn better atomistic representations from existing labeled data. Specifically, one of the most popular self-supervised learning methods in NLP (Devlin et al., 2019) and CV (He et al., 2022) is training a denoising autoencoder (Vincent et al., 2008), where the idea is to mask or corrupt a part of the input data and learn to reconstruct the original, uncorrupted data. Denoising assumes we know a unique target structure to denoise to – e.g., a sentence and an image in the case of NLP and CV. Indeed, this is the case for equilibrium structures (e.g., $S_{eq}$ at a local energy minimum in Figure 1b) as has been demonstrated by previous works leveraging denoising

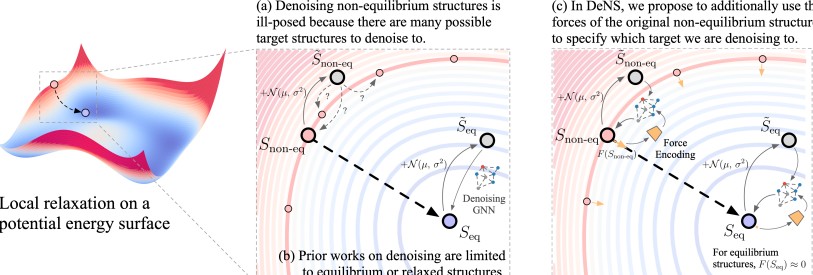

Figure 1: Illustration of denoising equilibrium and non-equilibrium structures. In this figure, we relax a non-equilibrium structure (red point) and form a local relaxation trajectory (black dotted arrow). All the points along the trajectory except the blue point are non-equilibrium structures.

for pretraining on atomistic data (Jiao et al., 2022; Zaidi et al., 2023; Liu et al., 2023; Wang et al., 2023; Feng et al., 2023a). However, most previous works are limited to equilibrium structures, and equilibrium structures constitute a small portion of available data since structures along a trajectory to get to a local minimum are all non-equilibrium. Hence, there is a need to generalize denoising to leverage the larger set of non-equilibrium structures.

Since a non-equilibrium structure has atom-wise forces and atoms are not confined to energy local minima, it has more possible atomic positions than an equilibrium one. As shown in Figure 1a, this can make denoising an ill-posed problem since there are many possible target structures to denoise to. To address the issue, we propose to take the forces of the original non-equilibrium structure as inputs when denoising non-equilibrium structures. Intuitively, the forces constraint atomic positions of a non-equilibrium structure. With the additional information, we are able to predict the original non-equilibrium structure satisfying the input forces instead of predicting any arbitrary structures as shown in Figure 1c. Previous works on denoising equilibrium structures (Jiao et al., 2022; Zaidi et al., 2023; Liu et al., 2023; Feng et al., 2023b;a) end up being a special case where the forces of original structures are close to zeros.

Based on the insight, in this paper, we propose to use **de**noising **n**on-equilibrium **s**tructures (**DeNS**) as an auxiliary task to better leverage atomistic data. For training DeNS, we first corrupt a structure by adding noise to its 3D atomic coordinates and then reconstruct the original uncorrupted structure by predicting the noise. For noise predictions, a model is given the forces of the original uncorrupted structure as inputs to make the transformation from a corrupted non-equilibrium structure to a uncorrupted non-equilibrium structure tractable. When used along with original tasks like predicting energy and forces of non-equilibrium structures, DeNS improves the performance of the original tasks with a marginal increase in training cost. We further discuss how DeNS can leverage more from training data and the connection to self-supervised learning methods in other domains.

Because DeNS requires encoding forces, it favors equivariant networks. They build up equivariant features at each node with vector spaces of irreducible representations (irreps) and have interactions or message passing between nodes with equivariant operations like tensor products. Since forces can be projected to vector spaces of irreps with spherical harmonics, equivariant networks can easily incorporate forces in node embeddings. Moreover, with the reduced complexity of equivariant operations (Passaro & Zitnick, 2023) and incorporating Transformer network design (Liao & Smidt, 2023; Liao et al., 2023) from NLP (Vaswani et al., 2017) and CV (Dosovitskiy et al., 2021), equivariant networks have become the state-of-the-art methods on large-scale atomistic datasets.

We conduct extensive experiments on OC20 (Chanussot* et al., 2021), OC22 (Tran* et al., 2022) and MD17 (Chmiela et al., 2017; Schütt et al., 2017; Chmiela et al., 2018) datasets and focus on how DeNS can improve the performance of equivariant networks. EquiformerV2 trained with DeNS achieves better S2EF results and comparable IS2RE results on OC20. EquiformerV2 trained with DeNS sets new state-of-the-art results on OC22. EquiformerV1 ($L_{max} = 2$) (Liao & Smidt, 2023) trained with DeNS achieves better results on MD17 than EquiformerV1 ($L_{max} = 3$) without DeNS and saves $3.1\times$ training time. DeNS can improve other equivariant networks like eSCN (Passaro & Zitnick, 2023) on OC20 and SEGNN-like networks (Brandstetter et al., 2022) on MD17.

## 2 RELATED WORKS

**Denoising 3D Atomistic Structures.**  Denoising structures have been used to boost the performance of GNNs on 3D atomistic datasets (Godwin et al., 2022; Jiao et al., 2022; Zaidi et al., 2023; Liu

et al., 2023; Feng et al., 2023b; Wang et al., 2023; Feng et al., 2023a). The approach is to first corrupt data by adding noise and then train a denoising autoencoder to reconstruct the original data by predicting the noise, and the motivation is that learning to reconstruct data enables learning generalizable representations (Devlin et al., 2019; He et al., 2022; Godwin et al., 2022; Zaidi et al., 2023). Since denoising equilibrium structures do not require labels and is self-supervised, similar to BERT (Devlin et al., 2019) and MAE (He et al., 2022), it is common to pre-train via denoising on a large dataset of equilibrium structures like PCQM4Mv2 (Nakata & Shimazaki, 2017) and then fine-tune with supervised learning on smaller downstream datasets. Besides, Noisy Nodes (Godwin et al., 2022) use denoising equilibrium structures as an auxiliary task along with original tasks without pre-training on another larger dataset. However, most of the previous works are limited to equilibrium structures, which occupy a much smaller amount of data than non-equilibrium ones. In contrast, the proposed DeNS generalizes denoising to non-equilibrium structures with force encoding so that we can improve the performance on the larger set of non-equilibrium structures. We provide a detailed comparison to previous works on denoising in Section A.2.

***SE(3)/E(3)-Equivariant Networks.*** Refer to Section A.1 for discussion on equivariant networks.

## 3 METHOD

### 3.1 PROBLEM SETUP

Calculating quantum mechanical properties like energy and forces of 3D atomistic systems is fundamental to many applications. An atomistic system can be one or more molecules, a crystalline material and so on. Specifically, each system $S$ is an example in a dataset and can be described as $S = \{(z_i, \mathbf{p}_i) \mid i \in \{1, ..., |S|\}\}$, where $z_i \in \mathbb{N}$ denotes the atomic number of the $i$-th atom and $\mathbf{p}_i \in \mathbb{R}^3$ denotes the 3D atomic position. The energy of $S$ is denoted as $E(S) \in \mathbb{R}$, and the atom-wise forces are denoted as $F(S) = \{\mathbf{f}_i \in \mathbb{R}^3 \mid i \in \{1, ..., |S|\}\}$, where $\mathbf{f}_i$ is the force acting on the $i$-th atom. In this paper, we define a system to be an equilibrium structure if all of its atom-wise forces are close to zeros. Otherwise, we refer to it as a non-equilibrium structure. Since non-equilibrium structures have non-zero atomic forces and thus are not at an energy minimum, they have more degrees of freedom and constitute a much larger set of possible structures than those at equilibrium.

In this work, we focus on the task of predicting energy and forces given non-equilibrium structures. Specifically, given a non-equilibrium structure $S_{\text{non-eq}}$, GNNs predict energy $\hat{E}(S_{\text{non-eq}})$ and atom-wise forces $\hat{F}(S_{\text{non-eq}}) = \left\{\hat{\mathbf{f}}_i(S_{\text{non-eq}}) \in \mathbb{R}^3 \mid i \in \{1, ..., |S_{\text{non-eq}}|\}\right\}$ and minimize the loss function:

$$\lambda_E \cdot \mathcal{L}_E + \lambda_F \cdot \mathcal{L}_F = \lambda_E \cdot |E'(S_{\text{non-eq}}) - \hat{E}(S_{\text{non-eq}})| + \lambda_F \cdot \frac{1}{|S_{\text{non-eq}}|} \sum_{i=1}^{|S_{\text{non-eq}}|} |\mathbf{f}'_i(S_{\text{non-eq}}) - \hat{\mathbf{f}}_i(S_{\text{non-eq}})|^2$$

(1)

$\lambda_E$ and $\lambda_F$ are energy and force coefficients controlling the relative importance between energy and force predictions. $E'(S_{\text{non-eq}}) = \frac{E(S_{\text{non-eq}}) - \mu_E}{\sigma_E}$ is the normalized ground truth energy obtained by first subtracting the original energy $E(S_{\text{non-eq}})$ by the mean of energy labels in the training set $\mu_E$ and then dividing by the standard deviation of energy labels $\sigma_E$. Similarly, $\mathbf{f}'_i = \frac{\mathbf{f}_i}{\sigma_F}$ is the normalized atom-wise force. For force predictions, we can either directly predict them from latent representations like node embeddings as commonly used for OC20 and OC22 datasets or take the negative gradients of predicted energy with respect to atomic positions for datasets like MD17.

### 3.2 DENOISING NON-EQUILIBRIUM STRUCTURES (DENS)

#### 3.2.1 FORMULATION OF DENOISING

Denoising structures have been used to improve the performance of GNNs on 3D atomistic datasets. They first corrupt data by adding noise and then train a denoising autoencoder to reconstruct the original data by predicting the noise. Specifically, given a 3D atomistic system $S = \{(z_i, \mathbf{p}_i) \mid i \in \{1, ..., |S|\}\}$, we create a corrupted structure $\tilde{S}$ by adding Gaussian noise with a tuneable standard deviation $\sigma$ to the atomic positions $\mathbf{p}_i$ of the original structure $S$:

$$\tilde{S} = \{(z_i, \tilde{\mathbf{p}}_i) \mid i \in \{1, ..., |S|\}\}, \quad \text{where} \quad \tilde{\mathbf{p}}_i = \mathbf{p}_i + \boldsymbol{\epsilon}_i \quad \text{and} \quad \boldsymbol{\epsilon}_i \sim \mathcal{N}(0, \sigma I_3) \tag{2}$$

We denote the set of noise added to $S$ as $\text{Noise}(S, \tilde{S}) = \{\boldsymbol{\epsilon}_i \in \mathbb{R}^3 \mid i \in \{1, ..., |S|\}\}$. When training a denoising autoencoder, GNNs take $\tilde{S}$ as inputs, output atom-wise noise predictions $\hat{\boldsymbol{\epsilon}}(\tilde{S})_i$ and minimize the L2 difference between normalized noise $\frac{\boldsymbol{\epsilon}_i}{\sigma}$ and noise predictions $\hat{\boldsymbol{\epsilon}}(\tilde{S})_i$:

$$\mathbb{E}_{p(S,\tilde{S})}\left[\frac{1}{|S|}\sum_{i=1}^{|S|}\left|\frac{\boldsymbol{\epsilon}_i}{\sigma}-\hat{\boldsymbol{\epsilon}}(\tilde{S})_i\right|^2\right] \tag{3}$$

$p(S,\tilde{S})$ denotes the probability of obtaining the corrupted structure $\tilde{S}$ from the original structure $S$. We divide the noise $\boldsymbol{\epsilon}_i$ by the standard deviation $\sigma$ to normalize the outputs of noise predictions.

When the original structure $S$ is an equilibrium structure, denoising is to find the structure corresponding to the nearest energy local minima given a high-energy corrupted structure. This makes denoising equilibrium structures a many-to-one mapping and a well-defined problem. However, when the original structure $S$ is a non-equilibrium structure, denoising, the transformation from a corrupted non-equilibrium structure to the original non-equilibrium one, can be an ill-posed problem since there are many possible target structures to denoise to as shown in Figure 1a.

### 3.2.2 FORCE ENCODING

The reason that denoising non-equilibrium structures can be ill-posed is because we do not provide sufficient information to specify the properties of the target structures. Concretely, given an original non-equilibrium structure $S_{\text{non-eq}}$ and its corrupted counterpart $\tilde{S}_{\text{non-eq}}$, some structures interpolated between $S_{\text{non-eq}}$ and $\tilde{S}_{\text{non-eq}}$ could be in the same data distribution and therefore be the potential target structures of denoising. In contrast, when denoising equilibrium structures as shown in Figure 1b, we implicitly provide the extra information that the target structure should be at equilibrium with near-zero forces, and this therefore limits the possibility of the target of denoising. Motivated by the assumption that the forces of the original structures are close to zeros when denoising equilibrium ones, we propose to encode the forces of original non-equilibrium structures when denoising non-equilibrium ones as illustrated in Figure 1c. Specifically, when training **de**noising **n**on-equilibrium **s**tructures (DeNS), GNNs take both a corrupted non-equilibrium structure $\tilde{S}_{\text{non-eq}}$ and the forces $F(S_{\text{non-eq}})$ of the original non-equilibrium structure $S_{\text{non-eq}}$ as inputs, output atom-wise noise predictions $\hat{\boldsymbol{\epsilon}}\left(\tilde{S}_{\text{non-eq}}, F(S_{\text{non-eq}})\right)_i$ and minimize the L2 difference between normalized noise $\frac{\boldsymbol{\epsilon}_i}{\sigma}$ and noise predictions $\hat{\boldsymbol{\epsilon}}\left(\tilde{S}_{\text{non-eq}}, F(S_{\text{non-eq}})\right)_i$:

$$\mathcal{L}_{\text{DeNS}} = \mathbb{E}_{p(S_{\text{non-eq}},\tilde{S}_{\text{non-eq}})}\left[\frac{1}{|S_{\text{non-eq}}|}\sum_{i=1}^{|S_{\text{non-eq}}|}\left|\frac{\boldsymbol{\epsilon}_i}{\sigma}-\hat{\boldsymbol{\epsilon}}\left(\tilde{S}_{\text{non-eq}}, F(S_{\text{non-eq}})\right)_i\right|^2\right] \tag{4}$$

Equation 4 is more general and reduces to Equation 3 when the target of denoising becomes equilibrium structures. Since we train GNNs with $\tilde{S}_{\text{non-eq}}$ and $F(S_{\text{non-eq}})$ as inputs and $\text{Noise}(S_{\text{non-eq}}, \tilde{S}_{\text{non-eq}})$ as outputs, they implicitly learn to leverage $F(S_{\text{non-eq}})$ to reconstruct $S_{\text{non-eq}}$ instead of predicting any arbitrary non-equilibrium structures. Comparing Index 1 and Index 2 in Table 1e, force encoding enables DeNS to significantly improve the performance.

Since DeNS requires encoding of forces, DeNS favors equivariant networks, which can easily incorporate forces as well as other higher-degree tensors into their node embeddings. Specifically, the node embeddings of equivariant networks are equivariant irreps features built from vectors spaces of irreducible representations (irreps) and contain $C_L$ channels of type-$L$ vectors with degree $L$ being in the range from 0 to maximum degree $L_{max}$. $C_L$ and $L_{max}$ are architectural hyper-parameters of equivariant networks. To obtain the force embedding $x_{\mathbf{f}}$ from the input force $\mathbf{f}$, we first project $\mathbf{f}$ into type-$L$ vectors, with $0 \leqslant L \leqslant L_{max}$, with spherical harmonics $Y^{(L)}\left(\frac{\mathbf{f}}{||\mathbf{f}||}\right)$ and then expand the number of channels from 1 to $C_L$ with an $SO(3)$ linear layer (Geiger et al., 2022; Geiger & Smidt, 2022):

$$x_{\mathbf{f}}^{(L)} = \text{SO3\_Linear}^{(L)}\left(||\mathbf{f}|| \cdot Y^{(L)}\left(\frac{\mathbf{f}}{||\mathbf{f}||}\right)\right) \tag{5}$$

$x_{\mathbf{f}}^{(L)}$ denotes the channels of type-$L$ vectors in force embedding $x_{\mathbf{f}}$, and $\text{SO3\_Linear}^{(L)}$ denotes the $SO(3)$ linear operation on type-$L$ vectors. Since we normalize the input force when using spherical harmonics, we multiply $Y^{(L)}\left(\frac{\mathbf{f}}{||\mathbf{f}||}\right)$ with the norm of input force $||\mathbf{f}||$ to recover the information of force magnitude. After computing force embeddings for all atom-wise forces, we add the force embeddings to initial node embeddings to encode forces in equivariant networks.

On the other hand, it might not be that intuitive to encode forces in invariant networks since their internal latent representations such as node embeddings and edge embeddings are scalars not geometric tensors. One potential manner of encoding forces in latent representations is to project them

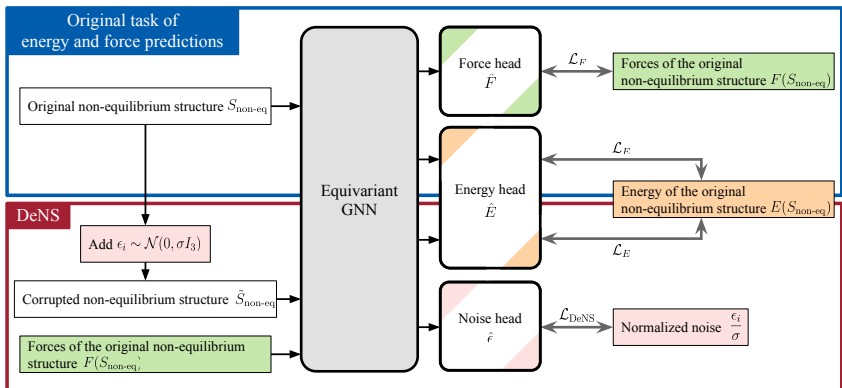

Figure 2: Training process when incorporating DeNS as an auxiliary task. The upper blue block corresponds to the original task of energy and force predictions (Equation 1), and the lower red block corresponds to training DeNS (Equation 6). "Equivariant GNN" and "energy head" are shared across the two tasks. For each batch of structures, we use the original task for some structures and DeNS for the others.

into edge embeddings by taking inner products between forces and edge vectors of relative positions. This process is the inverse of how GemNet-OC (Gasteiger et al., 2022) decodes forces from latent representations. Since equivariant networks have been shown to outperform invariant networks on datasets containing non-equilibrium structures and are simpler to encode forces, in this work, we mainly focus on equivariant networks and how DeNS can further advance their performance.

### 3.2.3 TRAINING DeNS

**Auxiliary Task.** We propose to train DeNS as an auxiliary task along with the original task of predicting energy and forces to improve the performance of energy and force predictions and summarize the training process in Figure 2. Specifically, given a batch of structures, for each structure, we decide whether we optimize the objective of DeNS (Equation 6) or the objective of the original task (Equation 1). This introduces an additional hyper-parameter $p_{\text{DeNS}}$, the probability of optimizing DeNS. We use an additional noise head for noise predictions, which slightly increases training time. Additionally, when training DeNS, similar to Noisy Nodes (Godwin et al., 2022), we also leverage energy labels and predict the energy of original structures. Therefore, given an original non-equilibrium structure $S_{\text{non-eq}}$ and the corrupted counterpart $\tilde{S}_{\text{non-eq}}$, training DeNS corresponds to minimizing the following loss function:

$$\lambda_E \cdot \mathcal{L}_E + \lambda_{\text{DeNS}} \cdot \mathcal{L}_{\text{DeNS}} = \lambda_E \cdot \left| E'(S_{\text{non-eq}}) - \hat{E}\left(\tilde{S}_{\text{non-eq}}, F(S_{\text{non-eq}})\right) \right| + \lambda_{\text{DeNS}} \cdot \mathcal{L}_{\text{DeNS}} \quad (6)$$

where $\mathcal{L}_{\text{DeNS}}$ denotes the loss function of denoising as defined in Equation 4. We note that we also encode forces as discussed in Section 3.2.2 to predict the energy of $S_{\text{non-eq}}$, and we share the energy prediction head across Equation 1 and Equation 6. The loss function introduces another hyper-parameter $\lambda_{\text{DeNS}}$, DeNS coefficient, controlling the importance of the auxiliary task. Besides, the process of corrupting structures also results in another hyper-parameter $\sigma$ as shown in Equation 2. We provide the pseudocode in Section E. We note that we only train DeNS with force encoding on the training set without using any force labels on the validation and testing sets.

**Multi-Scale Noise.** Inspired by prior works on denoising score matching (Song & Ermon, 2019; 2020), we empirically find that incorporating multiple noise scales together for denoising improves energy and force predictions on OC20 and OC22 datasets. Specifically, we choose the standard deviations $\{\sigma_k\}_{k=1}^T$ to be a geometric progression that satisfy $\frac{\sigma_T}{\sigma_{T-1}} = ... = \frac{\sigma_3}{\sigma_2} = \frac{\sigma_2}{\sigma_1} > 1$:

$$\sigma_k = \exp\left(\log_e \sigma_{\text{low}} + \frac{k-1}{T-1} \cdot (\log_e \sigma_{\text{high}} - \log_e \sigma_{\text{low}})\right) \quad \text{where} \quad k = 1, ..., T \quad (7)$$

Here we use $\sigma_1 = \sigma_{\text{low}} = 0.01$ and $T = 50$ and only tune $\sigma_T = \sigma_{\text{high}}$ when multi-scale noise is adopted. When training with DeNS, for each structure, we first sample a single noise standard deviation $\sigma$ from the $T$ values, and then follow Equations 2 and 4. We surmise that multi-scale noises are more likely to span the distribution of meaningful non-equilibrium structures across a diverse range of atom types and geometries compared to a fixed $\sigma$.

### 3.2.4 DISCUSSION

**How DeNS Can Improve Performance.** DeNS enables leveraging more from training data to improve the performance in the following two manners. First, DeNS adds noise to non-equilibrium

structures to generate structures with new geometries and therefore naturally achieves data augmentation (Godwin et al., 2022). Second, training DeNS enourages learning a different yet highly correlated interaction. Since we encode forces as inputs and predict the original structures in terms of noise corrections, DeNS enables learning the interaction of transforming forces into structures, which is the inverse of force predictions. As demonstrated in the works of Noisy Nodes (Godwin et al., 2022) and UL2 (Tay et al., 2023), training a single model with multiple correlated objectives to learn different interactions can help the performance on the original task.

**Connection to Self-Supervised Learning.**  DeNS shares similar intuitions to self-supervised learning methods like BERT (Devlin et al., 2019) and MAE (He et al., 2022) and other denoising methods (Vincent et al., 2008; 2010; Godwin et al., 2022; Zaidi et al., 2023) – they remove or corrupt a portion of input data and then learn to predict the original data. Learning to reconstruct data can help learning generalizable representations, and therefore these methods can use the task of reconstruction to improve the performance of downstream tasks. However, since DeNS requires force labels for encoding, DeNS does not belong to self-supervised learning strictly but instead provides another way to learn more from data. Therefore, we propose to use DeNS as an auxiliary task optimized along with original tasks and do not follow the previous practice of first pre-training and then fine-tuning. Additionally, we note that before obtaining a single equilibrium structure, we need to run relaxations and generate many intermediate non-equilibrium ones, which is the labeling process as well.  We hope that the ability to leverage more from non-equilibrium structures as proposed in this work can encourage researchers to release data containing intermediate non-equilibrium structures in addition to final equilibrium ones. Moreover, we note that DeNS can be used in fine-tuning. For example, we can first pre-train models on PCQM4Mv2 dataset and then fine-tune them on the smaller MD17 dataset with both the original task and DeNS.

**Marginal Increase in Training Time.**  Since we use an additional noise head for denoising, training with DeNS marginally increases the time of each training iteration. We optimize DeNS for some structures and the original task for the others for each training iteration, and we demonstrate that DeNS can improve the performance given the same amount of training iterations. Therefore, training with DeNS only marginally increase the overall training time.

## 4 EXPERIMENTS

### 4.1 OC20 DATASET

**Dataset and Tasks.**  We start with experiments on the large and diverse Open Catalyst 2020 dataset (OC20) (Chanussot* et al., 2021), which consists of about 1.2M Density Functional Theory (DFT) relaxation trajectories. Each DFT trajectory in OC20 starts from an initial structure of an adsorbate molecule placed on a catalyst surface, which is then relaxed with the revised Perdew-Burke-Ernzerhof (RPBE) functional (Hammer et al., 1999) calculations to a local energy minimum. Relevant to DeNS, all the intermediate structures from these trajectories, except the relaxed structure, are considered non-equilibrium structures. The relaxed or equilibrium structure has forces close to zero.

The primary task in OC20 is Structure to Energy and Forces (S2EF), which is to predict the energy and per-atom forces given an equilibrium or non-equilibrium structure from any point in the trajectory. These predictions are evaluated on energy and force mean absolute error (MAE). Once a model is trained for S2EF, it is used to run structural relaxations from an initial structure using the predicted forces till a local energy minimum is found. The energy predictions of these relaxed structures are evaluated on the Initial Structure to Relaxed Energy (IS2RE) task.

**Training Details.**  Please refer to Section B.1 for details on DeNS, architectures, hyper-parameters and training time.

#### 4.1.1 ABLATION STUDIES

We use EquiformerV2 (Liao et al., 2023) and S2EF-2M split of OC20 to investigate how DeNS-related hyper-parameters affect the performance, compare the results of training with and without DeNS and verify some design choices of DeNS.

**Hyperparameters.**  In Tables 1a, 1b, 1c, we vary $\sigma_{\text{high}}$, the upper bound on standard deviations of Gaussian noise, $p_{\text{DeNS}}$, the probability of optimizing DeNS, and $\lambda_{\text{DeNS}}$, the loss coefficient for DeNS, to study how the hyper-parameters of DeNS affect performance. We find that the optimal settings are similar when training for different epochs and have the following observations. First, as we increase $\sigma_{\text{high}}$, force predictions become worse monotonically, while energy predictions improve

| Epochs | $\sigma_{\text{high}}$ | forces | energy |
|---|---|---|---|
| 12 | 0.125 | 19.10 | 284 |
| 12 | 0.250 | 19.11 | 276 |
| 12 | 0.500 | 19.31 | 273 |
| 12 | 0.750 | 19.43 | 273 |
| 12 | 1.000 | 19.52 | 274 |
| 20 | 0.250 | 18.41 | 271 |
| 20 | 0.500 | 18.48 | 263 |
| 20 | 0.750 | 18.66 | 263 |
| 20 | 1.000 | 18.66 | 261 |
| 30 | 0.250 | 17.85 | 265 |
| 30 | 0.500 | 17.96 | 255 |
| 30 | 0.750 | 18.06 | 255 |
| 30 | 1.000 | 18.06 | 256 |

(a) Upper bound on standard deviations of Gaussian noise $\sigma_{\text{high}}$.

| Epochs | $p_{\text{DeNS}}$ | forces | energy |
|---|---|---|---|
| 12 | 0.125 | 19.64 | 276 |
| 12 | 0.250 | 19.31 | 273 |
| 12 | 0.500 | 19.32 | 271 |
| 12 | 0.750 | 19.90 | 281 |
| 20 | 0.250 | 18.66 | 263 |
| 20 | 0.500 | 18.49 | 262 |
| 20 | 0.750 | 18.92 | 269 |
| 30 | 0.250 | 18.06 | 255 |
| 30 | 0.500 | 17.83 | 255 |
| 30 | 0.750 | 18.24 | 262 |

(b) Probability of optimizing DeNS $p_{\text{DeNS}}$.

| Epochs | $\lambda_{\text{DeNS}}$ | forces | energy |
|---|---|---|---|
| 12 | 5 | 19.64 | 277 |
| 12 | 10 | 19.31 | 273 |
| 12 | 15 | 19.25 | 275 |
| 20 | 5 | 19.02 | 262 |
| 20 | 10 | 18.66 | 263 |
| 20 | 15 | 18.56 | 263 |
| 30 | 5 | 18.40 | 260 |
| 30 | 10 | 18.06 | 255 |
| 30 | 15 | 17.94 | 258 |

(c) Loss coefficient for DeNS $\lambda_{\text{DeNS}}$.

| | EquiformerV2 | | | | EquiformerV2 + DeNS | | | |
|---|---|---|---|---|---|---|---|---|
| Epochs | forces | energy | # params | training time | forces | energy | # params | training time |
| 12 | 20.46 | 285 | 83M | 1398 | 19.32 | 271 | 89M | 1501 |
| 20 | 19.78 | 280 | 83M | 2330 | 18.49 | 262 | 89M | 2501 |
| 30 | 19.42 | 278 | 83M | 3495 | 17.83 | 255 | 89M | 3752 |
| | eSCN | | | | eSCN + DeNS | | | |
| 20 | 20.61 | 290 | 52M | 1802 | 19.07 | 279 | 52M | 1829 |

(d) Comparison of training with and without DeNS.

| Index | | forces | energy |
|---|---|---|---|
| 1 | DeNS | 19.32 | 271 |
| 2 | Without force encoding | 20.16 | 271 |
| 3 | $\lambda_E = 0$ in Eq. 6 | 19.57 | 281 |
| 4 | Fixed $\sigma = 0.1$ | 19.58 | 283 |
| 5 | With $\sigma$ encoding | 19.66 | 279 |

(e) Design Choices.

Table 1: Ablation results of EquifomerV2 trained with DeNS on the 2M subset of the OC20 S2EF dataset. We report mean absolute errors for forces in meV/Å and energy in meV, and lower is better. Errors are averaged over the four validation sub-splits of OC20. The training time is in GPU-hours and measured on V100 GPUs. (a)-(c) We use $\sigma_{\text{high}} = 0.5$ for 12 epochs, $\sigma_{\text{high}} = 0.75$ for 20 and 30 epochs, $p_{\text{DeNS}} = 0.25$, and $\lambda_{\text{DeNS}} = 10$ as the default DeNS-related hyper-parameters and vary them to study how they affect the performance. The settings with the best energy-force trade-offs are marked in gray. (d) We train EquiformerV2 and eSCN and compare the results of training with and without DeNS. The results of EquiformerV2 are from (b). (e) We train EquiformerV2 for 12 epochs with the best setting in (b) to verify the design choices of DeNS.

but saturate at $\sigma_{\text{high}} = 0.5$. Second, $p_{\text{DeNS}} = 0.5$ and $\lambda_{\text{DeNS}} = 10$ and $15$ work better than any other values across the three different epochs.

**Comparison of Training with and without DeNS.** Table 1d summarizes the results of training with and without DeNS. For EquiformerV2, incorporating DeNS as an auxiliary task boosts the performance of energy and force predictions, and only increases training time by $7.4\%$ and the number of parameters from 83M to 89M. Particularly, EquiformerV2 trained with DeNS for 12 epochs outperforms EquiformerV2 trained without DeNS for 30 epochs, saving $2.3\times$ training time. Additionally, we also show that DeNS can be applicable to other equivariant networks like eSCN, and training eSCN with DeNS improves both energy and force MAE while slightly increasing training time by $1.5\%$. The different amount of increase in training time between EquiformerV2 and eSCN is because they use different modules for noise predictions.

**Design Choices.** We conduct experiments to verify the design choices of DeNS and summarize the results in Table 1e. All the models follow the best setting of training with DeNS for 12 epochs in Table 1b. Comparing Index 1 and Index 2, we show that encoding forces $F(S_{\text{non-eq}})$ in Equations 4 and 6 enables denoising non-equilibrium structures to further improve performance. DeNS without force encoding only results in slightly better force MAE than training without DeNS as in Table 1d. We also compare DeNS with and without force encoding on the MD17 dataset in Section D.3 and find that force encoding is critical. Comparing Index 1 and Index 3, we demonstrate that predicting energy of original structures given corrupted ones can be helpful to the original task. Additionally, we compare the performance of using a fixed $\sigma$ and multi-scale noise, and the comparison between Index 1 and Index 4 shows that multi-scale noise improves both energy and force predictions. Since we sample standard deviation $\sigma$ when using multi-scale noise, we also investigate whether we need to encode $\sigma$. The comparison between Index 1 and Index 5 shows that DeNS without $\sigma$ encoding works better, and thus we can use the same approach when we use either a fixed $\sigma$ or multi-scale noise.

### 4.1.2 MAIN RESULTS

**All + MD.** We train EquiformerV2 (160M) with DeNS on the S2EF-All+MD split of OC20. The model follows the same configuration as EquiformerV2 (153M) trained without DeNS, and the

| Model | Throughput Samples / GPU sec. ↑ | S2EF validation Energy MAE (meV) ↓ | Force MAE (meV/Å) ↓ | S2EF test Energy MAE (meV) ↓ | Force MAE (meV/Å) ↓ | IS2RE test Energy MAE (meV) ↓ |
|---|---|---|---|---|---|---|
| GemNet-OC-L-E (Gasteiger et al., 2022) | 7.5 | 239 | 22.1 | 230 | 21.0 | - |
| GemNet-OC-L-F (Gasteiger et al., 2022) | 3.2 | 252 | 20.0 | 241 | 19.0 | - |
| GemNet-OC-L-F+E (Gasteiger et al., 2022) | - | - | - | - | - | 348 |
| SCN L=6 K=16 (4-tap 2-band) (Zitnick et al., 2022) | - | - | - | 228 | 17.8 | 328 |
| SCN L=8 K=20 (Zitnick et al., 2022) | - | - | - | 237 | 17.2 | 321 |
| eSCN L=6 K=20 (Passaro & Zitnick, 2023) | 2.9 | 243 | 17.1 | 236 | 16.2 | 323 |
| EquiformerV2 ($\lambda_E = 4$, 31M) (Liao et al., 2023) | 7.1 | 232 | 16.3 | 228 | 15.5 | 315 |
| EquiformerV2 ($\lambda_E = 2$, 153M) (Liao et al., 2023) | 1.8 | 230 | 14.6 | 227 | 13.8 | 311 |
| EquiformerV2 ($\lambda_E = 4$, 153M) (Liao et al., 2023) | 1.8 | 227 | 15.0 | 219 | 14.2 | 309 |
| EquiformerV2 + DeNS ($\lambda_E = 4$, 160M) | 1.8 | **221** | **14.2** | **216** | **13.4** | **308** |

Table 2: OC20 results on S2EF validation and test splits and IS2RE test split when trained on OC20 S2EF-All+MD split. Throughput is reported as the number of structures processed per GPU-second during training and measured on V100 GPUs.

| Model | Number of parameters | S2EF-Total validation Energy MAE (meV) ↓ ID | OOD | Force MAE (meV/Å) ↓ ID | OOD | S2EF-Total test Energy MAE (meV) ↓ ID | OOD | Force MAE (meV/Å) ↓ ID | OOD | IS2RE-Total test Energy MAE (meV) ↓ ID | OOD |
|---|---|---|---|---|---|---|---|---|---|---|---|
| GemNet-OC (Gasteiger et al., 2022) | 39M | 545 | 1011 | 30 | 40 | 374 | 829 | 29.4 | 39.6 | 1329 | 1584 |
| GemNet-OC (trained on OC20 + OC22) (Gasteiger et al., 2022) | 39M | 464 | 859 | 27 | 34 | 311 | 689 | 26.9 | 34.2 | 1200 | 1534 |
| eSCN (Passaro & Zitnick, 2023) | 200M | - | - | - | - | 350 | 789 | 23.8 | 35.7 | - | - |
| EquiformerV2 ($\lambda_E = 1, \lambda_F = 1$) (Liao et al., 2023) | 122M | **343** | 580 | 24.42 | 34.31 | **182.8** | 677.4 | 22.98 | 35.57 | 1084 | 1444 |
| EquiformerV2 ($\lambda_E = 4, \lambda_F = 100$) (Liao et al., 2023) | 122M | 433 | 629 | 22.88 | **30.70** | 263.7 | 659.8 | 21.58 | 32.65 | 1119 | 1440 |
| EquiformerV2 + DeNS ($\lambda_E = 4, \lambda_F = 100$) | 127M | 395 | **532** | **22.76** | 31.53 | 226.9 | **619.2** | **21.30** | **32.42** | **1029** | **1392** |

Table 3: OC22 results on S2EF-Total validation and test splits and IS2RE-Total test split. Models are trained on the OC22 S2EF-Total training split unless otherwise noted.

additional parameters are due to force encoding and one additional equivariant graph attention for noise predictions. We report results in Table 2. All test results are computed via the EvalAI evaluation server[1]. EquiformerV2 trained with DeNS achieves better S2EF results and comparable IS2RE results, setting the new state-of-the-art results. The improvement is not as significant as that on OC20 S2EF-2M and MD17 (Section 4.3) datasets since the OC20 S2EF-All+MD training set contains much more structures along relaxation trajectories, making new 3D geometries generated by DeNS less helpful. However, DeNS is valuable because most datasets are not as large as OC20 S2EF-All+MD dataset but have sizes closer to OC20 S2EF-2M and MD17 datasets.

## 4.2 OC22 DATASET

**Dataset and Tasks.** The Open Catalyst 2022 (OC22) dataset (Tran* et al., 2022) focuses on oxide electrocatalysis and consists of about 62k DFT relaxations obtained with Perdew-Burke-Ernzerhof (PBE) functional calculations. One crucial difference in OC22, compared to OC20, is that the energy targets in OC22 are DFT total energies. DFT total energies are harder to predict but are the most general and closest to a DFT surrogate, offering the flexibility to study property prediction beyond adsorption energies. Analogous to the task definitions in OC20, the primary tasks in OC22 are S2EF-Total and IS2RE-Total. We train models on the OC22 S2EF-Total dataset, which has $8.2M$ structures, and evaluate them on energy and force MAE on the S2EF-total validation and test splits. After that, we use these models to perform structure relaxations starting from initial structures in the IS2RE-Total test split and evaluate the predicted relaxed energies on energy MAE.

**Training Details.** Please refer to Section C.1 for details on architectures, hyper-parameters and training time.

**Results.** First, we conduct ablation studies to investigate the effects of DeNS-related hyper-parameters in Section C.2. Second, we use the best hyper-parameter setting in Section C.2 to train EquiformerV2 of 18 blocks with DeNS and report the results in Table 3. Compared to EquiformerV2 trained with different energy and force coefficients but without DeNS, EquiformerV2 trained with DeNS improves the trade-offs between energy and force MAE, achieving comparable energy MAE to EquiformerV2 ($\lambda_E = 1$, $\lambda_F = 1$) trained without DeNS and overall better force MAE than EquiformerV2 ($\lambda_E = 4$, $\lambda_F = 100$) trained without DeNS. For IS2RE-Total, EquiformerV2 trained with DeNS achieves the best energy MAE results. The improvement on IS2RE-Total from training with DeNS on only OC22 is comparable to that of training on the much larger OC20 and OC22 datasets in previous works. Specifically, training GemNet-OC on OC20 and OC22 datasets (about 138M + 8.4M structures) improves IS2RE-Total energy MAE ID by 129meV and OOD by 50meV compared to training GemNet-OC on only OC22 dataset (8.4M structures). Compared to training without DeNS, training EquiformerV2 with DeNS improves ID by 90meV and OOD by 48meV. Thus, training with DeNS clearly improves sample efficiency and performance on OC22.

---

[1] `eval.ai/web/challenges/challenge-page/712`

| | Aspirin | | Benzene | | Ethanol | | Malonaldehyde | | Naphthalene | | Salicylic acid | | Toluene | | Uracil | | Training time | Number of |
|---|---|---|---|---|---|---|---|---|---|---|---|---|---|---|---|---|---|---|
| Model | energy | forces | energy | forces | energy | forces | energy | forces | energy | forces | energy | forces | energy | forces | energy | forces | (GPU-hours)↓ | parameters |
| SchNet (Schütt et al., 2017) | 16.0 | 58.5 | 3.5 | 13.4 | 3.5 | 16.9 | 5.6 | 28.6 | 6.9 | 25.2 | 8.7 | 36.9 | 5.2 | 24.7 | 6.1 | 24.3 | - | - |
| DimeNet (Gasteiger et al., 2020) | 8.8 | 21.6 | 3.4 | 8.1 | 2.8 | 10.0 | 4.5 | 16.6 | 5.3 | 9.3 | 5.8 | 16.2 | 4.4 | 9.4 | 5.0 | 13.1 | - | - |
| PaiNN (Schütt et al., 2021) | 6.9 | 14.7 | - | - | 2.7 | 9.7 | 3.9 | 13.8 | 5.0 | 3.3 | 4.9 | 8.5 | 4.1 | 4.1 | 4.5 | 6.0 | - | - |
| TorchMD-NET (Thölke & Fabritiis, 2022) | 5.3 | 11.0 | 2.5 | 8.5 | 2.3 | 4.7 | 3.3 | 7.3 | **3.7** | 2.6 | **4.0** | 5.6 | **3.2** | 2.9 | **4.1** | 4.1 | - | - |
| NequIP ($L_{max} = 3$) (Batzner et al., 2022) | 5.7 | 8.0 | - | - | **2.2** | 3.1 | 3.3 | 5.6 | 4.9 | **1.7** | 4.6 | **3.9** | 4.0 | 2.0 | 4.5 | **3.3** | - | - |
| Equiformer ($L_{max} = 2$) | 5.3 | 7.2 | **2.2** | 6.6 | **2.2** | 3.1 | 3.3 | 5.8 | **3.7** | 2.1 | 4.5 | 4.1 | 3.8 | 2.1 | 4.3 | **3.3** | 17 | 3.50M |
| Equiformer ($L_{max} = 3$) | 5.3 | 6.6 | 2.5 | 8.1 | **2.2** | 2.9 | **3.2** | 5.4 | 4.4 | 2.0 | 4.3 | **3.9** | 3.7 | 2.1 | 4.3 | 3.4 | 59 | 5.50M |
| Equiformer ($L_{max} = 2$) + DeNS | **5.1** | **5.7** | 2.3 | **6.1** | **2.2** | **2.6** | **3.2** | **4.4** | **3.7** | **1.7** | 4.3 | **3.9** | 3.5 | **1.9** | 4.2 | **3.3** | 19 | 4.00M |

Table 4: Mean absolute error results on the MD17 testing set. Energy and force are in units of meV and meV/Å. We additionally report the time of training different Equiformer models averaged over all molecules and the numbers of parameters to show that the proposed DeNS can improve performance with minimal overhead.

| | Method | | | Aspirin | | Benzene | | Ethanol | | Malonaldehyde | | Naphthalene | | Salicylic acid | | Toluene | | Uracil | |
|---|---|---|---|---|---|---|---|---|---|---|---|---|---|---|---|---|---|---|---|
| Index | Attention | Layer normalization | DeNS | energy | forces | energy | forces | energy | forces | energy | forces | energy | forces | energy | forces | energy | forces | energy | forces |
| 1 | ✓ | ✓ | | 5.3 | 7.2 | **2.2** | 6.6 | **2.2** | 3.1 | 3.3 | 5.8 | **3.7** | 2.1 | 4.5 | 4.1 | 3.8 | 2.1 | 4.3 | **3.3** |
| 2 | ✓ | ✓ | ✓ | **5.1** | **5.7** | 2.3 | **6.1** | **2.2** | **2.6** | **3.2** | **4.4** | **3.7** | **1.7** | 4.3 | **3.9** | 3.5 | **1.9** | 4.2 | **3.3** |
| 3 | | ✓ | | 5.2 | 7.7 | 2.4 | 6.2 | 2.3 | 3.9 | 3.3 | 6.2 | 3.8 | 2.2 | **4.1** | 4.7 | **3.3** | 2.4 | 4.2 | 4.4 |
| 4 | | ✓ | ✓ | 5.2 | 6.1 | 2.4 | **6.1** | **2.2** | 2.9 | **3.2** | 5.1 | **3.7** | **1.7** | 4.2 | **3.9** | 3.4 | 2.0 | 4.2 | 3.4 |
| 5 | | | | 5.3 | 9.3 | 2.4 | 9.2 | 2.3 | 4.5 | 3.4 | 8.2 | **3.7** | 2.4 | 4.2 | 5.5 | **3.3** | 2.9 | 4.2 | 4.8 |
| 6 | | | ✓ | 5.2 | 7.3 | 2.4 | 8.1 | **2.2** | 3.4 | 3.4 | 6.7 | **3.7** | 1.9 | 4.2 | 4.4 | 3.4 | 2.2 | 4.2 | 3.8 |

Table 5: Mean absolute error results of variants of Equiformer ($L_{max} = 2$) without attention and layer normalization on the MD17 testing set. Energy and force are in units of meV and meV/Å. Index 1 and Index 2 correspond to "Equiformer ($L_{max} = 2$)" and "Equiformer ($L_{max} = 2$) + DeNS" in Table 4.

### 4.3 MD17 Dataset

**Dataset.** The MD17 dataset (Chmiela et al., 2017; Schütt et al., 2017; Chmiela et al., 2018) consists of molecular dynamics simulations of small organic molecules. The task is to predict the energy and forces of these non-equilibrium molecules. We use $950$ and $50$ different configurations for training and validation sets and the rest for the testing set.

**Training Details.** Please refer to Section D.2 for additinoal implementation details of DeNS, hyper-parameters and training time.

**Results.** We train Equiformer ($L_{max} = 2$) (Liao & Smidt, 2023) with DeNS based on their implementation and summarize the results in Table 4. DeNS improves the results on all molecules, and Equiformer ($L_{max}$ = 2) trained with DeNS achieves overall best results. Particularly, Equiformer ($L_{max} = 2$) trained with DeNS acheives better results on all the tasks and requires $3.1\times$ less training time than Equiformer ($L_{max} = 3$) trained without DeNS. This demonstrates that for this small dataset, training an auxiliary task and using data augmentation are more efficient and result in larger performance gain than increasing $L_{max}$ from 2 to 3. Besides, training with DeNS marginally increase the training time and the number of parameters since we have one additional equivariant graph attention for noise predictions. Additionally, we find that the gains from training DeNS as an auxiliary task are comparable to pre-training. Zaidi et al. (2023) uses TorchMD-NET (Thölke & Fabritiis, 2022) pre-trained on the PCQM4Mv2 dataset and report results on Aspirin. Their improvement on force MAE is about $17.2\%$ (Table 3 in Zaidi et al. (2023)). Training Equiformer with DeNS results in $20.8\%$ improvement on force MAE without relying on another dataset. Note that we only increase training time by $10.5\%$ while their method takes much more time since PCQM4Mv2 dataset is $3000\times$ larger than MD17. Moreover, we also train variants of Equiformer ($L_{max} = 2$) by removing attention and layer normalization to investigate the performance gain of DeNS on different network archtiectures. The results are summarized in Table 5, and DeNS improves all the models. We note that Equiformer without attention and layer normalization reduces to SEGNN (Brandstetter et al., 2022) but with a better radial basis function. Since the models cover many variants of equivariant networks, this suggests that DeNS is general and can be helpful to many equivariant networks.

### 5 Conclusion

In this paper, we propose to use **de**noising **n**on-equilibrium **s**tructures (DeNS) as an auxiliary task to better leverage training data and improve performance of original tasks of energy and force predictions. Denoising non-equilibrium structures can be an ill-posed problem since there are many possible target structures to denoise to. To address the issue, we propose to take the forces of original structures as inputs to specify which non-equilibrium structures we are denoising. With force encoding, DeNS successfully improve the performance of original tasks when it is used as an auxiliary task. We conduct extensive experiments on OC20, OC22 and MD17 datasets to demonstrate DeNS can boost the performance of energy and force predictions with minimal increase in training cost and are applicable to many equivariant networks.

## 6 REPRODUCIBILITY STATEMENT

We include details on DeNS, architectures, hyper-parameters and training time in Sec. B.1 (OC20), Sec. C.1 (OC22) and Sec. D (MD17).

We submit our code reproducing the results of EquiformerV2 trained with DeNS on OC20 S2EF-2M dataset and Equiformer ($L_{max} = 2$) trained with DeNS on MD17 dataset. Following the author guide, after the discussion forums are opened for all submitted papers, we will make a comment directed to the reviewers and area chairs and put a link to an anonymous repository.

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

APPENDIX

## A  RELATED WORKS

### A.1  *SE(3)/E(3)*-EQUIVARIANT NETWORKS

Equivariant neural networks (Thomas et al., 2018; Kondor et al., 2018; Weiler et al., 2018; Fuchs et al., 2020; Miller et al., 2020; Townshend et al., 2020; Batzner et al., 2022; Jing et al., 2021; Schütt et al., 2021; Satorras et al., 2021; Brandstetter et al., 2022; Thölke & Fabritiis, 2022; Le et al., 2022; Musaelian et al., 2022; Batatia et al., 2022; Liao & Smidt, 2023; Passaro & Zitnick, 2023; Liao et al., 2023) use vector spaces of irreducible representations (irreps) for equivariant irreps features and adopt equivariant operations such as tensor products to achieve equivariance to 3D rotation. Previous works differ in which equivariant operations are used and the combination of those operations. TFN (Thomas et al., 2018) and NequIP (Batzner et al., 2022) use tensor products for equivariant graph convolution with linear messages, with the latter utilizing extra gate activation (Weiler et al., 2018). SEGNN (Brandstetter et al., 2022) applies gate activation to messages passing for non-linear messages (Gilmer et al., 2017; Sanchez-Gonzalez et al., 2020). SE(3)-Transformer (Fuchs et al., 2020) adopts equivariant dot product attention with linear messages. Equiformer (Liao & Smidt, 2023) improves upon previous models by combining MLP attention and non-linear messages and additionally introducing equivariant layer normalization and regularizations like dropout (Srivastava et al., 2014) and stochastic depth (Huang et al., 2016). However, these networks rely on compute-intensive $SO(3)$ convolutions built from tensor products, and therefore they can only use small values for maximum degrees $L_{max}$ of irreps features. eSCN (Passaro & Zitnick, 2023) significantly reduces the complexity of $SO(3)$ convolutions by first rotating irreps features based on relative positions and then applying $SO(2)$ linear layers, enabling higher values of $L_{max}$. EquiformerV2 (Liao et al., 2023) adopts eSCN convolutions and proposes an improved version of Equiformer to better leverage the power of higher $L_{max}$, achieving the current state-of-the-art results on OC20 (Chanussot* et al., 2021) and OC22 (Tran* et al., 2022) datasets.

We refer readers to the works (Liao & Smidt, 2023; Liao et al., 2023) for detailed background on equivariant networks.

### A.2  COMPARISON TO PREVIOUS WORKS ON DENOISING

We discuss the comparisons between previous works on denoising (Godwin et al., 2022; Zaidi et al., 2023; Feng et al., 2023b; Wang et al., 2023) and this work in chronological order as below.

Godwin et al. (2022) first proposes the idea of adding noise to 3D coordinates and then using denoising as an auxiliary task. The auxiliary task is trained along with the original task without relying on another large dataset. Their approach requires known equilibrium structures and therefore

is limited to QM9 (Ramakrishnan et al., 2014; Ruddigkeit et al., 2012) and OC20 IS2RE datasets and can not be applied to force prediction. For QM9, all the structures are at equilibrium, and for OC20 IS2RE, the target of denoising is the equilibrium relaxed structure. Denoising without force encoding is well-defined on both QM9 and OC20 IS2RE. In contrast, this work proposes using force encoding to generalize their approach to force prediction and non-equilibrium structures, which have much larger datasets than equilibrium ones. Force encoding can achieve better results on OC20 S2EF dataset without any overhead (Index 1 and Index 2 in Table 1(e)) and is necessary on MD17 dataset (Section D.3).

Zaidi et al. (2023) adopts the denoising approach proposed by Godwin et al. (2022) as a pre-training method and therefore requires another large dataset containing unlabelled equilibrium structures for pre-training. On the other hand, Godwin et al. (2022) and this work use denoising along with the original task and do not use any additional unlabeled data.

Feng et al. (2023b) follows the same practice of pre-training via denoising (Zaidi et al., 2023) and proposes a different manner of adding noise. Specifically, they separate noise into dihedral angle noise and coordinate noise and only learn to predict coordinate noise. However, dihedral angle noise requires tools like RDKit to obtain rotatable bonds and cannot be applied to other datasets like OC20 and OC22.

Although Zaidi et al. (2023) and Feng et al. (2023b) report results of force prediction on MD17 dataset, they first pre-train models on PCQM4Mv2 dataset (Nakata & Shimazaki, 2017) and then fine-tune the pre-trained models on MD17 dataset. We note that their setting is different from ours since we do not use any dataset for pre-training. As for fine-tuning on MD17 dataset, Zaidi et al. (2023) simply follows the same practice in standard training. Feng et al. (2023b) explores fine-tuning with objectives similar to Noisy Nodes (Godwin et al., 2022), but the performance gain is much smaller than ours. Concretely, in Table 5 in Feng et al. (2023b), the improvement on force prediction on Aspirin is about 2.6% while we improve force MAE by 20.8%.

Wang et al. (2023) uses the same pre-training method as Zaidi et al. (2023) but applies it to ANI-1 (Smith et al., 2017) and ANI-1x (Smith et al., 2018) datasets, which contain non-equilibrium structures. However, Wang et al. (2023) does not encode forces, and denoising non-equilibrium structures without force encoding can sometimes lead to worse results compared to training without denoising.

# B  DETAILS OF EXPERIMENTS ON OC20

## B.1  TRAINING DETAILS

Since each structure in OC20 S2EF dataset has pre-defined fixed atoms with zero forces and free atoms with non-zero forces, when training DeNS, we only add noise to free atoms and denoise corrupted free atoms. Additionally, we find that multi-scale noise works better than using a fixed $\sigma$ and thus corrupt structures with multi-scale noise when training DeNS.

We add an additional equivariant graph attention to EquiformerV2 for noise predictions. We mainly follow the hyper-parameters of training EquiformerV2 without DeNS on OC20 S2EF-2M and S2EF-All+MD datasets. For training EquiformerV2 on OC20 S2EF-All+MD dataset, we increase the number of epochs from 1 to 2 for better performance. This results in higher training time than other methods. However, we note that we already demonstrate training with DeNS can achieve better results given the same amount of training time in Table 1d. Table 6 summarizes the hyper-parameters of training EquiformerV2 with DeNS for the ablation studies on OC20 S2EF-2M dataset in Section 4.1.1 and for the main results on OC20 S2EF-All+MD datasets in Section 4.1.2. Please refer to the work of EquiformerV2 (Liao et al., 2023) for details of the architecture.

V100 GPUs with 32GB are used to train models. We use 16 GPUs for training on OC20 S2EF-2M dataset and 128 GPUs for OC20 S2EF-All+MD dataset. The training time and the numbers of parameters of different models on OC20 S2EF-2M dataset can be found in Table 1d. For OC20 S2EF-All+MD dataset, the training time is 44281 GPU-hours and the number of parameters is 160M.

| Hyper-parameters | S2EF-2M | EquiformerV2 (160M) on S2EF-All+MD |
|---|---|---|
| Optimizer | AdamW | AdamW |
| Learning rate scheduling | Cosine learning rate with linear warmup | Cosine learning rate with linear warmup |
| Warmup epochs | 0.1 | 0.01 |
| Maximum learning rate | $2 \times 10^{-4}$ for 12 epochs | $4 \times 10^{-4}$ |
| | $4 \times 10^{-4}$ for 20, 30 epochs | |
| Batch size | 64 | 512 |
| Number of epochs | 12, 20, 30 | 2 |
| Weight decay | $1 \times 10^{-3}$ | $1 \times 10^{-3}$ |
| Dropout rate | 0.1 | 0.1 |
| Stochastic depth | 0.05 | 0.1 |
| Energy coefficient $\lambda_E$ | 2 | 4 |
| Force coefficient $\lambda_F$ | 100 | 100 |
| Gradient clipping norm threshold | 100 | 100 |
| Model EMA decay | 0.999 | 0.999 |
| Cutoff radius (Å) | 12 | 12 |
| Maximum number of neighbors | 20 | 20 |
| Number of radial bases | 600 | 600 |
| Dimension of hidden scalar features in radial functions $d_{edge}$ | $(0, 128)$ | $(0, 128)$ |
| Maximum degree $L_{max}$ | 6 | 6 |
| Maximum order $M_{max}$ | 2 | 3 |
| Number of Transformer blocks | 12 | 20 |
| Embedding dimension $d_{embed}$ | $(6, 128)$ | $(6, 128)$ |
| $f_{ij}^{(L)}$ dimension $d_{attn\_hidden}$ | $(6, 64)$ | $(6, 64)$ |
| Number of attention heads $h$ | 8 | 8 |
| $f_{ij}^{(0)}$ dimension $d_{attn\_alpha}$ | $(0, 64)$ | $(0, 64)$ |
| Value dimension $d_{attn\_value}$ | $(6, 16)$ | $(6, 16)$ |
| Hidden dimension in feed forward networks $d_{ffn}$ | $(6, 128)$ | $(6, 128)$ |
| Resolution of point samples $R$ | 18 | 18 |
| Upper bound on standard deviations of Gaussian noise $\sigma_{high}$ | - | 0.75 |
| Probability of optimizing DeNS $p_{DeNS}$ | - | 0.25 |
| Loss coefficient for DeNS $\lambda_{DeNS}$ | - | 10 |

Table 6: Hyper-parameters of training EquiformerV2 with DeNS on OC20 S2EF-2M dataset and OC20 S2EF-All+MD dataset. For OC20 S2EF-2M dataset, the DeNS-related hyper-parameters, $\sigma_{high}$, $p_{DeNS}$ and $\lambda_{DeNS}$, can be found in Table 1.

## C DETAILS OF EXPERIMENTS ON OC22

### C.1 TRAINING DETAILS

Different from OC20, all the atoms in a structure in OC22 are free. Therefore, we add noise to and denoise all atoms when training DeNS. We follow the practice on OC20 dataset and use multi-scale noise to corrupt structures.

We add an additional equivariant graph attention to EquiformerV2 for noise predictions. Table 7 summarizes the hyper-parameters of ablation results on OC22 in Table 8. The DeNS-related hyper-parameters can be found in Table 8. As for the result in Table 3, we increase the number of Transformer blocks to 18, increase the batch size to 256 and train for 12 epochs. For DeNS-related hyper-parameters, we use $\sigma_{high} = 0.5$, $p_{DeNS} = 0.5$ and $\lambda_{DeNS} = 50$.

We use 32 V100 GPUs (32GB) for the ablation results, and the training time and the numbers of parameters can be found in Table 8. We use 64 V100 GPUs (32GB) for the result in Table 3. The training time is 10163 GPU-hours, and the number of parameters 127M.

### C.2 ABLATION RESULTS

We train EquiformerV2 of 12 blocks with DeNS to study how the hyper-parameters of DeNS affect the performance and compare with EquiformerV2 trained without DeNS. The results are summarized in Table 8. Compared to EquiformerV2 trained without DeNS, EquiformerV2 trained with DeNS achieves significantly better energy MAE and comparable force MAE. We find the best hyper-parameter setting is $\sigma_{high} = 0.5$, $p_{DeNS} = 0.5$ and $\lambda_{DeNS} = 50$.

| Hyper-parameters | Value or description |
|---|---|
| Optimizer | AdamW |
| Learning rate scheduling | Cosine learning rate with linear warmup |
| Warmup epochs | 0.1 |
| Maximum learning rate | $2 \times 10^{-4}$ |
| Batch size | 128 |
| Number of epochs | 6 |
| Weight decay | $1 \times 10^{-3}$ |
| Dropout rate | 0.1 |
| Stochastic depth | 0.1 |
| Energy coefficient $\lambda_E$ | 4 |
| Force coefficient $\lambda_F$ | 100 |
| Gradient clipping norm threshold | 50 |
| Model EMA decay | 0.999 |
| Cutoff radius (Å) | 12 |
| Maximum number of neighbors | 20 |
| Number of radial bases | 600 |
| Dimension of hidden scalar features in radial functions $d_{edge}$ | $(0, 128)$ |
| Maximum degree $L_{max}$ | 6 |
| Maximum order $M_{max}$ | 2 |
| Number of Transformer blocks | 12 |
| Embedding dimension $d_{embed}$ | $(6, 128)$ |
| $f_{ij}^{(L)}$ dimension $d_{attn\_hidden}$ | $(6, 64)$ |
| Number of attention heads $h$ | 8 |
| $f_{ij}^{(0)}$ dimension $d_{attn\_alpha}$ | $(0, 64)$ |
| Value dimension $d_{attn\_value}$ | $(6, 16)$ |
| Hidden dimension in feed forward networks $d_{ffn}$ | $(6, 128)$ |
| Resolution of point samples $R$ | 18 |

Table 7: Hyper-parameters for OC22 dataset.

| | # Parameters | Training time (GPU-hours) | $\sigma_{high}$ | $p_{DeNS}$ | $\lambda_{DeNS}$ | Energy MAE (meV) ↓ | | Force MAE (meV/Å) ↓ | |
|---|---|---|---|---|---|---|---|---|---|
| | | | | | | ID | OOD | ID | OOD |
| EquiformerV2 (Liao et al., 2023) | 83M | 3184 | - | - | - | 436 | 641 | 23.37 | 31.77 |
| EquiformerV2 + DeNS | 89M | 3446 | | | | | | | |
| | | | 0.25 | 0.25 | 50 | 415 | 590 | 23.13 | 31.32 |
| | | | 0.50 | 0.25 | 50 | 420 | 586 | 22.93 | **31.38** |
| | | | 0.75 | 0.25 | 50 | 419 | 623 | 23.25 | 32.04 |
| | | | 1.00 | 0.25 | 50 | 429 | 621 | 23.11 | 31.49 |
| | | | 0.50 | 0.50 | 50 | **408** | **579** | 23.02 | 31.41 |
| | | | 0.50 | 0.25 | 25 | 415 | 639 | 23.34 | 32.22 |
| | | | 0.50 | 0.25 | 75 | 429 | 588 | 23.02 | 31.74 |
| | | | 0.50 | 0.25 | 100 | 424 | 665 | **22.96** | 31.68 |

Table 8: Ablation results of training EquiformerV2 with DeNS on OC22. All models have 12 blocks and are trained on the OC22 S2EF-Total training split with energy coefficient $\lambda_E = 4$ and force coefficient $\lambda_F = 100$. Errors are evaluated on the validation split. The best hyperparameter settings are marked in gray.

# D   DETAILS OF EXPERIMENTS ON MD17

## D.1   ADDITIONAL DETAILS OF DENS

**Denoising Partially Corrupted Structures.** Empirically, we find that adding noise to all atoms in a structure can lead to limited performance gain of DeNS on the MD17 dataset, and we surmise that there are several structures satisfying input forces if we add noise to all atoms, making denoising a less well-defined problem. We note that the issue can depend on datasets, and adding noise to all atoms can still result in better performance on the OC20 and OC22 datasets. To address the issue, we propose to only add noise to a subset of atoms and denoise partially corrupted structures. Specifically, for corrupted atoms, we encode their atom-wise forces and predict the noise. For other uncorrupted atoms, we do not encode forces but predict them as in the original task. We also predict the energy of the original structures given partially corrupted structures. This solution introduces an extra hyper-parameter $r_{DeNS}$, the ratio of the number of corrupted atoms to that of all atoms. For OC20 and OC22 datasets, we add noises to all atoms when training DeNS, and therefore $r_{DeNS} = 1.0$.

**Implementation Details.** It is necessary that gradients consider both the original task and DeNS when updating learnable parameters, and this affects how we sample structures for DeNS when only a single GPU is used for training models on the MD17 dataset. We zero out forces corresponding to structures used for the original task so that a single forward-backward propagation can consider both DeNS and the original task. In contrast, if we switch between DeNS and the original task for different iterations, gradients only consider either DeNS or the original task, and we find that this does not result in better performance on the MD17 dataset than training without DeNS.

| Hyper-parameter | Aspirin | Benzene | Ethanol | Malonaldehyde | Naphthalene | Salicylic acid | Toluene | Uracil |
|---|---|---|---|---|---|---|---|---|
| Energy coefficient $\lambda_E$ | 1 | 1 | 1 | 1 | 2 | 1 | 1 | 1 |
| Force coefficient $\lambda_F$ | 80 | 80 | 80 | 100 | 20 | 80 | 80 | 20 |
| Probability of optimizing DeNS $p_{\text{DeNS}}$ | 0.25 | 0.25 | 0.25 | 0.25 | 0.25 | 0.125 | 0.125 | 0.25 |
| Standard deviation of Gaussian noises $\sigma$ | 0.05 | 0.05 | 0.05 | 0.05 | 0.05 | 0.025 | 0.025 | 0.05 |
| DeNS coefficient $\lambda_{\text{DeNS}}$ | 5 | 5 | 5 | 5 | 5 | 5 | 5 | 5 |
| Ratio of the number of corrupted atoms to that of all atoms $r_{\text{DeNS}}$ | 0.25 | 0.25 | 0.25 | 0.25 | 0.25 | 0.25 | 0.25 | 0.25 |

Table 9: Hyper-parameters of training with DeNS on the MD17 dataset. The other hyper-parameters not listed here are the same as the original Equiformer ($L_{max} = 2$) trained without DeNS.

| Index | Method | Aspirin | | Benzene | | Ethanol | | Malonaldehyde | | Naphthalene | | Salicylic acid | | Toluene | | Uracil | |
|---|---|---|---|---|---|---|---|---|---|---|---|---|---|---|---|---|---|
| | | energy | forces | energy | forces | energy | forces | energy | forces | energy | forces | energy | forces | energy | forces | energy | forces |
| 1 | Equiformer ($L_{max} = 2$) | 5.3 | 7.2 | **2.2** | 6.6 | **2.2** | 3.1 | 3.3 | 5.8 | **3.7** | 2.1 | 4.5 | 4.1 | 3.8 | 2.1 | 4.3 | **3.3** |
| 2 | Equiformer ($L_{max} = 2$) + DeNS without force encoding | 8.6 | 9.1 | 2.3 | 6.3 | 2.3 | 3.3 | **3.2** | 5.8 | 7.7 | 6.1 | 5.2 | 10.6 | 3.7 | 2.0 | 5.5 | 6.5 |
| 3 | Equiformer ($L_{max} = 2$) + DeNS with force encoding | **5.1** | **5.7** | 2.3 | **6.1** | **2.2** | **2.6** | **3.2** | **4.4** | **3.7** | **1.7** | **4.3** | **3.9** | **3.5** | **1.9** | **4.2** | **3.3** |

Table 10: Comparison between denoising with and without force encoding on MD17 dataset. Mean absolute error results are evaluated on the MD17 testing set. Energy and force are in units of meV and meV/Å. Index 1 and Index 3 correspond to "Equiformer ($L_{max} = 2$)" and "Equiformer ($L_{max} = 2$) + DeNS" in Table 4.

## D.2 TRAINING DETAILS

We use the same codebase as Equiformer (Liao & Smidt, 2023) for experiments on the MD17 dataset and follow most of the original hyper-parameters for training with DeNS. For training DeNS, we use an additional equivariant graph attention for noise predictions, which slightly increases training time and the number of parameters. We use single-scale noise with a fixed standard deviation $\sigma$ when corrupting structures. The hyper-parameters introduced by training DeNS and the values of energy coefficient $\lambda_E$ and force coefficient $\lambda_F$ on different molecules can be found in Table 9. Empirically, we find that linearly decaying DeNS coefficient $\lambda_{\text{DeNS}}$ to 0 thoughout the training can result in better performance. For the Equiformer variant without attention and layer normalization, we find that using normal distributions to initialize weights can result in training divergence and therefore we use uniform distributions. For some molecules, we find training Equiformer variant without attention and layer normalization with DeNS is unstable and therefore reduce the learning rate to $3 \times 10^{-4}$.

We use one A5000 GPU with 24GB to train different models for each molecule. We report the training time averaged over all molecules. Training Equiformer ($L_{max} = 2$) without DeNS takes about 17 hours, and training Equiformer ($L_{max} = 3$) without DeNS takes about 59 hours. When DeNS is used as an auxiliary task, training Equiformer ($L_{max} = 2$) takes 19 hours. As for numbers of parameters, Equiformer ($L_{max} = 2$) without DeNS has 3.50M parameters, Equiformer ($L_{max} = 2$) with DeNS has 4.00M parameters, and Equiformer ($L_{max} = 3$) without DeNS has 5.50M parameters.

## D.3 COMPARISON BETWEEN DENOISING WITH AND WITHOUT FORCE ENCODING

We compare DeNS with and without force encoding on MD17 to demonstrate that force encoding is critical. The results are summarized in Table 10. DeNS with force encoding (Index 3) achieves the best results across all molecules. Compared to training without denoising (Index 1), training with DeNS without force encoding (Index 2) only achieves slightly better results on some molecules (i.e., Benzene, Malondaldehyde, and Toluene) and much worse results on others. For molecules on which training DeNS without force encoding is helpful, adding force encoding can further achieve even better results. For other molecules, force encoding is critical for DeNS to be effective.

## E PSEUDOCODE FOR TRAINING WITH DENS

We provide the pseudocode for training with DeNS in Algorithm 1 and note that Line 5 can be parallelized. For denoising partially corrupted structures discussed in Section D.1, we only add noise to a subset of atoms (Line 14) and predict the corresponding noise (Line 21).

---

**Algorithm 1** Training with DeNS

1: **Input:**
 $p_{\text{DeNS}}$: probability of optimizing DeNS
 $\lambda_{\text{DeNS}}$: DeNS coefficient
 $\sigma$: standard deviation of Gaussian noise if multi-scale noise is not used
 $\sigma_{\text{high}}$: upper bound on standard deviations of Gaussian noise if multi-scale noise is used
 $\sigma_{\text{low}}$: lower bound on standard deviations of Gaussian noise if multi-scale noise is used
 $\lambda_E$: energy coefficient
 $\lambda_F$: force coefficient
 GNN: graph neural network for predicting energy, forces and noise
2: **while** training **do**
3:    $\mathcal{L}_{\text{total}} = 0$
4:    Sample a batch of $B$ structures $\{(S_{\text{non-eq}})^j \mid j \in \{1, ..., B\}\}$ from the training set
5:    **for** $j = 1$ to $B$ **do**                    ▷ This for loop can be parallelized
6:        Let $(S_{\text{non-eq}})^j = \left\{ (z_i, \mathbf{p}_i) \mid i \in \{1, ..., |(S_{\text{non-eq}})^j|\} \right\}$
7:        Sample $p$ from a uniform distribution $\mathbf{U}(0, 1)$ to determine whether to optimize DeNS
8:        **if** $p < p_{\text{DeNS}}$ **then**              ▷ Optimize DeNS based on Equation 6
9:            **if** multi-scale noise is used **then**
10:               Sample $\sigma_{\text{sample}}$ from $\{\sigma_k\}_{k=1}^T$
11:           **else**
12:               $\sigma_{\text{sample}} = \sigma$
13:           **end if**
14:           **for** $i = 1$ to $|(S_{\text{non-eq}})^j|$ **do**
15:               $\boldsymbol{\epsilon}_i \sim \mathcal{N}(0, \sigma_{\text{sample}} I_3)$
16:               $\tilde{\mathbf{p}}_i = \mathbf{p}_i + \boldsymbol{\epsilon}_i$
17:           **end for**
18:           Let $(\tilde{S}_{\text{non-eq}})^j = \left\{ (z_i, \tilde{\mathbf{p}}_i) \mid i \in \{1, ..., |(S_{\text{non-eq}})^j|\} \right\}$
19:           $\hat{E}, \_, \hat{\boldsymbol{\epsilon}} \leftarrow \text{GNN}\left( (\tilde{S}_{\text{non-eq}})^j, F\left((S_{\text{non-eq}})^j\right) \right)$
20:           $\mathcal{L}_E = \left| E'\left((S_{\text{non-eq}})^j\right) - \hat{E} \right|$
21:           $\mathcal{L}_{\text{DeNS}} = \frac{1}{|(S_{\text{non-eq}})^j|} \sum_{i=1}^{|(S_{\text{non-eq}})^j|} \left| \frac{\boldsymbol{\epsilon}_i}{\sigma_{\text{sample}}} - \hat{\boldsymbol{\epsilon}}_i \right|^2$  ▷ Calculate $\mathcal{L}_{\text{DeNS}}$ based on Equation 4
22:           $\mathcal{L}_{\text{total}} = \mathcal{L}_{\text{total}} + \lambda_E \cdot \mathcal{L}_E + \lambda_{\text{DeNS}} \cdot \mathcal{L}_{\text{DeNS}}$
23:       **else**                        ▷ Optimize the original task based on Equation 1
24:           $\hat{E}, \hat{F}, \_ \leftarrow \text{GNN}\left( (S_{\text{non-eq}})^j \right)$
25:           $\mathcal{L}_E = \left| E'\left((S_{\text{non-eq}})^j\right) - \hat{E} \right|$
26:           $\mathcal{L}_F = \frac{1}{|(S_{\text{non-eq}})^j|} \sum_{i=1}^{|(S_{\text{non-eq}})^j|} |\mathbf{f}_i'\left((S_{\text{non-eq}})^j\right) - \hat{\mathbf{f}}_i|^2$
27:           $\mathcal{L}_{\text{total}} = \mathcal{L}_{\text{total}} + \lambda_E \cdot \mathcal{L}_E + \lambda_F \cdot \mathcal{L}_F$
28:       **end if**
29:   **end for**
30:   $\mathcal{L}_{\text{total}} = \frac{\mathcal{L}_{\text{total}}}{B}$
31:   Optimize GNN based on $\mathcal{L}_{\text{total}}$
32: **end while**

---

## F  VISUALIZATION OF CORRUPTED STRUCTURES

We visualize how adding noise of different scales affect structures in OC20, OC22 and MD17 datasets in Figure 3, Figure 4 and Figure 5, respectively.

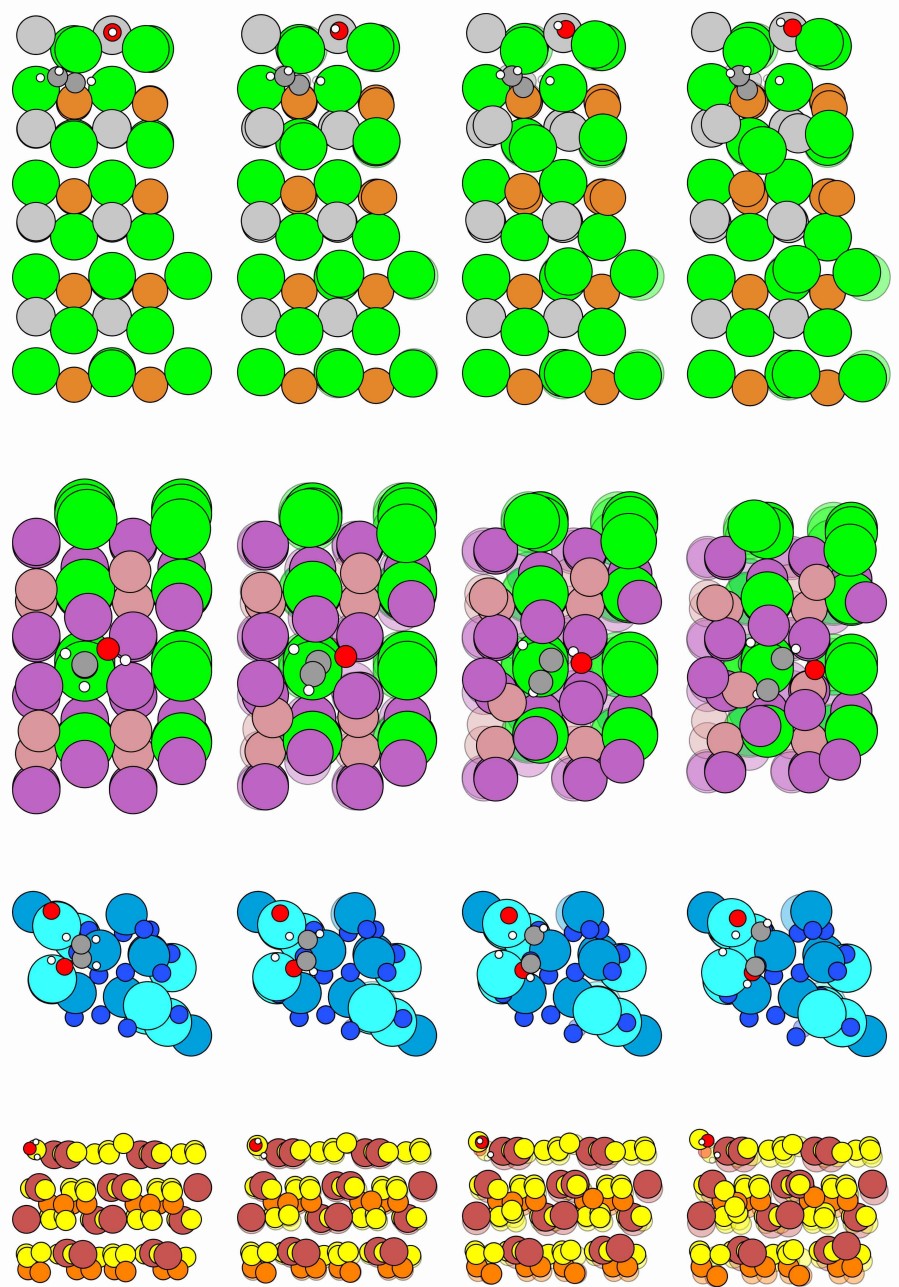

Figure 3: Visualization of corrupted structures in OC20 dataset. We add noise of different scales to original structures (column 1). For each row, we sample $\epsilon_i \sim \mathcal{N}(0, I_3)$, multiply $\epsilon_i$ with $\sigma = 0.1$ (column 2), $0.3$ (column 3) and $0.5$ (column 4), and add the scaled noise to the original structures. For columns 2, 3 and 4, the ligher colors denote the atomic positions of the original structures.

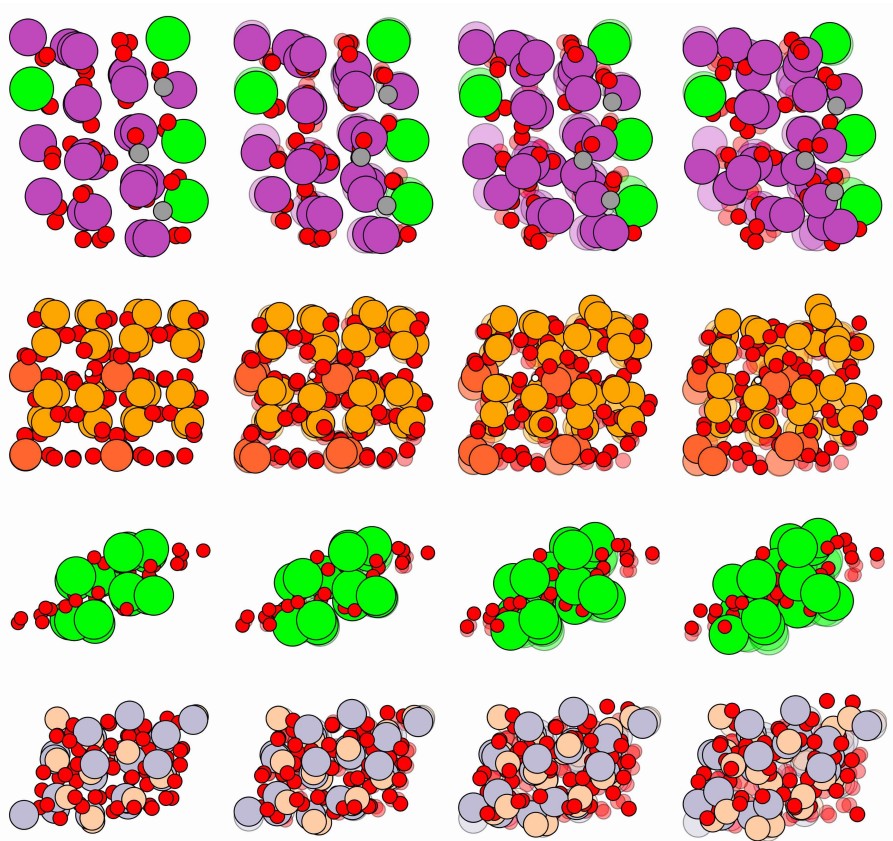

Figure 4: Visualization of corrupted structures in OC22 dataset. We add noise of different scales to original structures (column 1). For each row, we sample $\epsilon_i \sim \mathcal{N}(0, I_3)$, multiply $\epsilon_i$ with $\sigma = 0.1$ (column 2), $0.3$ (column 3) and $0.5$ (column 4), and add the scaled noise to the original structures. For columns 2, 3 and 4, the ligher colors denote the atomic positions of the original structures.

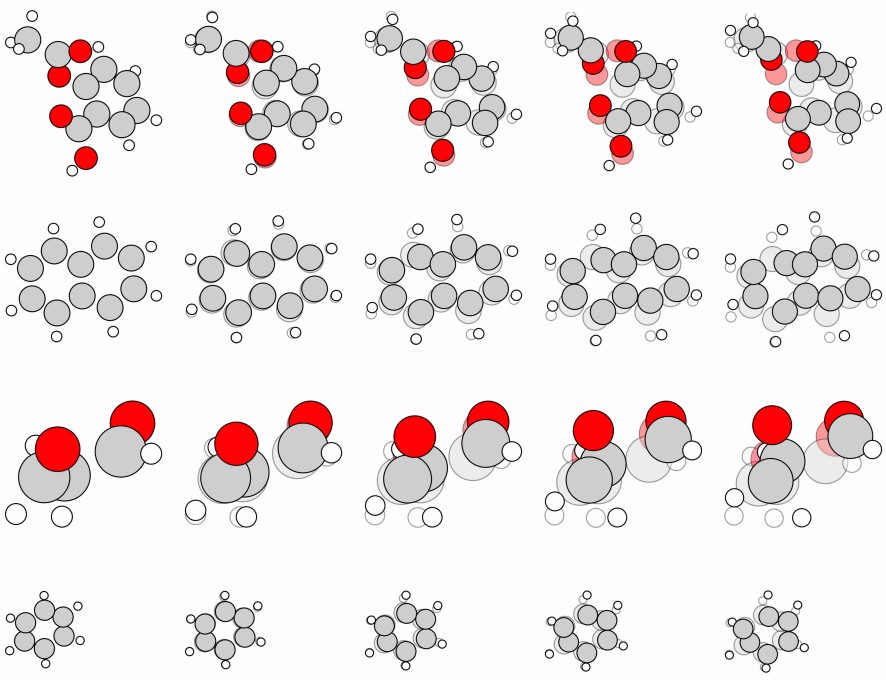

Figure 5: Visualization of corrupted structures in MD17 dataset. We add noise of different scales to original structures (column 1). For each row, we sample $\epsilon_i \sim \mathcal{N}(0, I_3)$, multiply $\epsilon_i$ with $\sigma = 0.01$ (column 2), $0.03$ (column 3), $0.05$ (column 4) and $0.07$ (column 5), and add the scaled noise to the original structures. For columns 2, 3, 4 and 5, the ligher colors denote the atomic positions of the original structures. Here we add noise to all the atoms in a structure for better visual effects.

