# OpenReview forum: "Generalizing Denoising to Non-Equilibrium Structures Improves Equivariant Force Fields"
_ICLR.cc/2024/Conference — Submitted to ICLR 2024_

### Official Review · Reviewer_NSpY · 2023-10-28

**Soundness:** 3 good
**Presentation:** 2 fair
**Contribution:** 4 excellent
**Rating:** 6
**Confidence:** 2

**Summary:**

This paper proposes a new auxiliary task to improve the training of networks on the prediction of the energy and forces from the structure of an atomistic system. It proposes to consider the recovery of the 3d structure after perturbing it with Gaussian noise (similar to a denoising autoencoder). Yet, since this plain (structure-to-structure) denoising problem is only well-posed for equilibrium structures, the authors propose to consider the forces of the original structure as an additional input along with a corrupted non-equilibrium structure, which seems to make the denoising problem well-posed again. The resulting ability to also consider  non-equilibrium structures significantly increases the available amount of training data and results in significant and systematic numerical improvements on several data sets.

**Strengths:**

I am not familiar with machine learning on atomistic systems, such that my evaluation has to be treated with some care. Yet,
- The idea seems to be novel, very well motivated by language processing and computer vision, and nicely resolves the ambiguity of denoising non-equilibrium structures with the help of additional input.
- The numerical results (on huge and apparently very competitive datasets) are very promising as they are largely able to improve the state-of-the-art.
- Ablation studies over hyper-parameters, architectural choices, and loss functions indicate a very well-designed method.

**Weaknesses:**

The presentation of the work could be improved significantly. While part of my difficulty in understanding the presentation is surely due to my lack of knowledge in the particular field, I believe some aspects hold for scientific texts in general
- Abbreviations should not be used before they are introduced. The abstract already refers to "S2EF" and "IS2RE results" (clarified in the numerical results), refers to network architectures without citation and even uses variables ($L_\max = 2$ and $L_\max=3$) whose meaning is assumed to be known.
- It is not explained why forces make the denoising problem on non-equilibrium structures well-posed and also not how the forces are obtained. Is it correct to assume that the potential of any structure can be computed and that the forces are the gradient thereof? If so, it would be just two additional sentences of explanation that make the work much more clear.
- The property "equivariant" is frequently used. While I know what it means, it was unclear to me what kind of equivariance (with respect to which transformations) is meant and why that influences the ability to encode forces. Rotations and translations of 3d coordinates/vectors?
- I do not know the term "type-L vectors" - what does it mean? I tried googling but did not have direct success. Thus, I'd recommend defining it. Also, what is "the projection of $f$ into type-L vectors with spherical harmonics"? Representing the function f with L-many coefficients corresponding to spherical harmonics basis functions?
- It would have helped me to cite something for "SO(3) linear layer".
- Please double-check the manuscript for typos.

Please be aware that I am not part of your main audience. Thus, feel free to adjust the writing for those aspects where you believe your main audience would also agree.

From my (quite uncertain) point of view, the strengths seem to clearly outweigh the suboptimal presentation (particularly if the latter is well suited for a more expert audience).

------
As an update after the discussion phase, the points I thought were strong (contribution+numerical evaluation) are rather seen as critical by reviewers who are much more familiar with the field. As my initial rating was based on the assumption of a significant novelty, such that I am lowering my score to account for this.

**Questions:**

In addition to some small things raised above, I have two further questions:
- It was strange to me to decide for a cost function for each batch during training with some probability. Will this not be equal to a weighted linear combination between the two terms in expectation?
- The ablation in Table 1 (b) indicates that $p_{DeNS}=0.5$ is the best, but also the largest tested value. Wouldn't it make sense to ablate values $>0.5$?

---

> ### Author Response · Authors · 2023-11-23
> **Response to Reviewer NSpY (1/2)**
>
> We thank the reviewer for helpful feedback and address the comments below.
>
> ---
>
> > 1. [Weakness 1] Abbreviations should not be used before they are introduced. The abstract already refers to "S2EF" and "IS2RE results" (clarified in the numerical results), refers to network architectures without citation and even uses variables (L_{max} = 2 and L_{max} = 3) whose meaning is assumed to be known.
>
> Thanks for pointing these out! We have updated the abstract.
>
> ---
>
> > 2. [Weakness 2] It is not explained why forces make the denoising problem on non-equilibrium structures well-posed and also not how the forces are obtained. Is it correct to assume that the potential of any structure can be computed and that the forces are the gradient thereof? If so, it would be just two additional sentences of explanation that make the work much more clear.
>
> The ground-truth force labels are obtained from DFT / MD calculations depending on the dataset and are from the training set.
>
> For predicting forces, as mentioned in Section 3.1, we either take the gradient of potential energy or directly predict them from node embeddings.
>
> Restating from the introduction and Section 3.2.2, denoising non-equilibrium structures without encoding forces as input is ill-posed because there are many possible target structures to denoise to. This is unlike the case for equilibrium structures which are at local minima, and so there exists a unique target structure to denoise to given perturbed structures. Encoding atomic forces in DeNS helps identify the unique non-equilibrium structure to denoise to. This is also illustrated in Figure 1. More specifically, since we train GNNs with $S_{\text{non-eq}}'$ and $F(S_{\text{non-eq}})$ as inputs and $\text{Noise}(S_{\text{non-eq}}, S_{\text{non-eq}}')$ as outputs, they implicitly learn to leverage $F(S_{\text{non-eq}})$ to reconstruct $S_{\text{non-eq}}$ instead of predicting any arbitrary non-equilibrium structures. (Here we use $S_{\text{non-eq}}'$ to denote corrupted non-equilibrium structures.)
>
> ---
>
> > 3. [Weakness 3] While I know what it means, it was unclear to me what kind of equivariance (with respect to which transformations) is meant and why that influences the ability to encode forces. Rotations and translations of 3d coordinates/vectors?
>
> Yes, the equivariance in this work refers to equivariance to 3D rotation and invariance to 3D translation, which is commonly implied in the context of learning force fields. Besides, we also discuss some equivariant networks and provide pointers to detailed background in Section A.1.
>
> As mentioned in Section 3.2.2, the node embeddings of equivariant networks consist of different type-$L$ vectors, where $L$ denotes degree, and forces belong to type-1 vectors. Therefore, to encode forces, we can simply add forces to the part of type-1 vectors in node embeddings.
>
> Please let us know if we can help clarify anything else.
>
> ---
>
> > 4. [Weakness 4] What are type-$L$ vectors? What is "the projection of f into type-$L$ vectors with spherical harmonics"?
>
> Type-$L$ vectors are (2$L$+1)-dimensional vectors that rotate $L$ times faster than 3D Euclidean vectors when we rotate the 3D coordinates. They can be obtained by taking tensor products of 3D relative positions (type-1 vectors).
>
> The projection of $f$ (type-1 vector) into type-$L$ vectors with spherical harmonics means that we use spherical harmonics $Y^{(L)}$ to transform type-1 vectors into type-$L$ vectors so that we can consider higher frequency (higher $L$).
>
> We note that the details can be found in [1] as mentioned in Section A.1.
>
> Reference:
> [1] Liao et al. Equiformer: Equivariant Graph Attention Transformer for 3D Atomistic Graphs. ICLR 2023.
>
>
> ---
>
> > 5.  [Weakness 5] Cite SO(3) linear layers.
>
> Thanks! We will add the citation of e3nn [1] after SO(3) linear layers.
>
> Reference:
> [1] Geiger et al. e3nn: Euclidean Neural Networks. ArXiv 2022.
>
>
> ---
>
> > 6. [Weakness 6] Double check the manuscript for typos.
>
> Thanks! We will fix them.
>
> ---
>
> > 7. [Question 1] It was strange to me to decide for a cost function for each batch during training with some probability. Will this not be equal to a weighted linear combination between the two terms in expectation?
>
> Yes, that should be equal to a weighted linear combination in expectation. However, the input structures corresponding to the two different loss functions (Equation 1 and Equation 6) are different. For Equation 1, we take the original structures as inputs while for Equation 6, we add noise and then take the corrupted structures as inputs. Therefore, we need to decide whether to add noise, and this introduces some probability and affects the subsequent loss function. We provide the pseudocode in Section E.

---

> > ### Author Response · Authors · 2023-11-23
> > **Response to Reviewer NSpY (2/2)**
> >
> > > 8. [Question 2] The ablation in Table 1 (b) indicates that $p_{\text{DeNS}}$ = 0.5 is the best, but also the largest tested value. Wouldn't it make sense to ablate values > 0.5?
> >
> > We added experiments with $p_{\text{DeNS}}$ = 0.75 for 12, 20 and 30 epochs in Table 1(b). $p_{\text{DeNS}}$ = 0.5 is still the best hyperparameter choice, but this makes Table 1(b) more complete. Thanks for the suggestion!

---

### Official Review · Reviewer_erXC · 2023-10-30

**Soundness:** 3 good
**Presentation:** 3 good
**Contribution:** 3 good
**Rating:** 5
**Confidence:** 3

**Summary:**

This paper proposes a method called denoising non-equilibrium structures (DeNS), which generalizes to a larger set of non-equilibrium structures without relying on additional datasets for pre-training, and the effectiveness of DeNS is demonstrated on the OC20, OC22, and MD17 datasets, achieving better results and faster training times compared to existing methods.

**Strengths:**

1. The inverse denoising framework presented in this work is quite interesting, as it proposes a novel possibility of combining structures, forces, and energy (or other properties) in a dual or inverse setting.
2. The experiments and ablation study conducted are robust and extensive, accompanied by meticulous analysis.
3. Well-written and easy to understand.

**Weaknesses:**

1. The results show limited improvements in the OC20, OC22, and MD17 datasets.
2. Maybe could add more results from the denoising framework with other backbones to provide a comprehensive understanding of its performance.
3. I believe that developing a general AI-based molecular dynamics (MD) method is more crucial than specifically designing and tuning for the OC dataset. A more generalized approach could be beneficial to the community by focusing on the broader application of deep learning-based MD methods across different systems, such as drug molecules, crsytal materials or polymers.

**Questions:**

1. If the improvements are not solely due to data augmentation, but rather are attributed to the inverse or dual denoising setting, it would be valuable to explore the generalization ability of the S2EF system in other systems.
2. The tunable standard deviation (σ) denoising strategy appears to be crucial, it might be beneficial to incorporate visualizations of σ during the training process for better illustration.

---

> ### Author Response · Authors · 2023-11-23
> **Response to  Reviewer erXC (1/1)**
>
> We thank the reviewer for helpful feedback and address the comments below.
>
> ---
>
> > 1. [Weakness 1] The results show limited improvements in the OC20, OC22, and MD17 datasets.
>
> Please see **General Response 3** (MD17), **General Response 5** (OC20) and **General Response 6** (OC22) for discussions on the performance gain.
>
> ---
>
> > 2. [Weakness 2] Maybe could add more results from the denoising framework with other backbones to provide a comprehensive understanding of its performance.
>
> We evaluate EquiformerV2 [1] and eSCN [2] on OC20 S2EF-2M dataset, Equiformer [3] and SEGNN [4] on MD17 dataset, and EquiformerV2 on OC22 dataset. These architectures improve the SoTA on each dataset. Of course more can always be done, but we believe 4 architectures spanning 3 datasets provides a sufficiently general evaluation of the proposed denoising approach. As a point of comparison, prior works cover less breadth in architectures – Noisy Nodes [5] only uses GNS [6], pre-training via denoising [7] uses GNS [6], GNS-TAT and TorchMD-NET [8], and fractional denoising [9] only conducts experiments with TorchMD-NET [8].
>
> Reference:
> [1] Liao et al. EquiformerV2: Improved Equivariant Transformer for Scaling to Higher-Degree Representations. ArXiv 2023.
> [2] Passaro et al. Reducing SO(3) Convolutions to SO(2) for Efficient Equivariant GNNs. ICML 2023.
> [3] Liao et al. Equiformer: Equivariant Graph Attention Transformer for 3D Atomistic Graphs. ICLR 2023.
> [4] Brandstetter et al. Geometric and Physical Quantities Improve E(3) Equivariant Message Passing. ICLR 2022.
> [5] Godwin et al. Simple GNN Regularisation for 3D Molecular Property Prediction and Beyond. ICLR 2022.
> [6] Sanchez-Gonzalez et al. Learning to simulate complex physics with graph networks. ICML 2020.
> [7] Zaidi et al. Pre-training via Denoising for Molecular Property Prediction. ICLR 2023.
> [8] Thölke et al. TorchMD-NET: Equivariant Transformers for Neural Network based Molecular Potentials. ICLR 2022.
> [9] Feng et al. Fractional Denoising for 3D Molecular Pre-training. ICML 2023.
>
> ---
>
> > 3. [Weakness 3] I believe that developing a general AI-based molecular dynamics (MD) method is more crucial than specifically designing and tuning for the OC dataset. A more generalized approach could be beneficial to the community by focusing on the broader application of deep learning-based MD methods across different systems, such as drug molecules, crystal materials or polymers.
>
> First, we show that DeNS improves performance on MD17 (see **General Response 3** and Table 4 in the paper).
>
> Second, we do not incorporate any constraints related to catalysis into DeNS, and therefore DeNS can be applied to all non-equilibrium structures, including those in MD simulations. From that perspective, DeNS is indeed generally applicable. Moreover, as mentioned in Section D.1 and D.2 (partial denoising, decaying the DeNS coefficient), DeNS can be specifically tuned for MD to further improve performance.
>
> Finally, now that we have a general method that seems to work well, we agree that pushing further on MD simulations across a broader range of chemistries (drug molecules, crystal materials, polymers) is the logical next step for future work!
>
> ---
>
> > 4. [Question 1] If the improvements are not solely due to data augmentation, but rather are attributed to the inverse or dual denoising setting, it would be valuable to explore the generalization ability of the S2EF system in other systems.
>
> The work of Noisy Nodes [1] has explored this. Comparing the first two rows in  Table 5 in [1], they find that simply adding data augmentation without denoising results in small performance gain, suggesting denoising is necessary. Besides, in Figure 3(B) and 3(C) in [1], they also show that adding noise as data augmentation results in strong regularization.
>
> Our validation and testing S2EF evaluations on OC20 and OC22 already test for generalizability since some of the splits involve out-of-distribution catalysts or adsorbates not seen during training.
>
> Reference:
> [1] Godwin et al. Simple GNN Regularisation for 3D Molecular Property Prediction and Beyond. ICLR 2022. https://arxiv.org/abs/2106.07971
>
> ---
>
> > 5. [Question 2] The tunable standard deviation (σ) denoising strategy appears to be crucial, and it might be beneficial to incorporate visualizations of σ during the training process for better illustration.
>
> Thanks for the suggestion! We visualize how structures in OC20, OC22 and MD17 datasets become after adding noise of different scales in Section F in the appendix of the revision.

---

### Official Review · Reviewer_AVUJ · 2023-10-30

**Soundness:** 2 fair
**Presentation:** 2 fair
**Contribution:** 3 good
**Rating:** 5
**Confidence:** 4

**Summary:**

The paper proposes an auxiliary task (not a pre-training) to help learn molecular tasks: denoising not only equilibrium (few are available) structures, but also denoising the not-yet-at-mechanical-equilibrium structures (much more numerous).

The key new idea is to provide also the forces (of the non-corrupted input) together with the corrupted structures, to make the problem well posed.

Rather heavy experiments show that the proposed method can sometimes beat the state of the art.

**Strengths:**

Originality:

The idea of the auxiliary task applied to non-equilibrium structures is great, here the paper allows to make this task well-defined by feeding the (uncorrupted) forces as input.


Clarity:

The paper explains carefully how the auxiliary task is added to the "default" model on which one wants to work.
I like a lot figure 1, which explains very clearly and concisely the idea.



Significance:

I think the paper makes the point for releasing more non-equilibrium structures (although, this is already the case in OC20 and OC22 if I understood well), which is important to state clearly to the community.  As authors say:

> We hope that the ability to leverage more from non-equilibrium structures as proposed in this work can encourage researchers to release data containing intermediate non-equilibrium structures in addition to final equilibrium ones.

**Weaknesses:**

Originality:

The original idea is not groundbreaking: auxiliary tasks are known, the specific case of denoising as well, here the novelty is only in feeding also the forces as input.

Quality:

it is not always clear from the results shown in tables, that the proposed method improves the SOTA significantly.
Also, since there are two measures of success (energy and forces), it's sometimes difficult to make a final decision.

Clarity:

The paper has some typos, but most importantly, experiments are discussed very quickly (too quickly). More space should be devoted to discuss results. For instance table 4 shows a nice improvement for DeNS on OOD splits (i.e. better generalization if I understand well that OOD is Out of Distribution as opposed to ID=In Distribution).

Significance:

Since the results are not strikingly better when using DeNS on the SOTA, and given it involves a number of additional hyperparameters (that obviously do not need very narrow fine-tuning, admitedly, but still, they involve more work and potential for problems), it is not clear yet whether the contribution would be used widely.

Table 1d is probably the most convincing result, to me (differences between models seem more significant). Is it computed on the validation set(s) ? (or train set ? I have a doubt).

In summary, this is red AI and the results are not significantly better than SOTA, thus not convincing for publication.

**Questions:**

I note that:
- trainings are very heavy.
- there is some hyper-param tuning, and DeNS is used on top of the best models (equiformerV2).
- energy and force: sometimes only one is better, but as you say, it's a balance.
- All that considered, most of the differences reported are not very significant. Maybe you could outline in green when a metric is significantly better in one model than in others (when this happens).
Can you answer and/or improve the discussions (and/or presentation of results) in the Experiments section, to show that indeed, the improvement is significant ?

If some tables show no significant improvement, it should be explained why.

If the method is mostly able to speed up training to achieve equivalent accuracy, state it (and it will weaken the paper's claim, but at the same time strengthen the submission).



Note:
Table 2 does not show a significant improvement from using DeNS




typos:
devication -> deviation
"L2 different" -> "L2 difference / L2 norm / squared difference "

**Details Of Ethics Concerns:**

Methodology:

Bad ecologial impact of this red AI-style research: for instance one of the experiment reports 3495hours of GPU of training time.

(Table I: The training time is in GPU-hours and measured on V100 GPUs.)

Again in Table 3: 3446 hours of GPU-time for training.

Overall, this research aims at shortening training time by using a smart data-augmentation strategy. But the results are very moderate, for an investment cost that is quite consequent.

But, at least, they do report these training times !

---

> ### Author Response · Authors · 2023-11-23
> **Response to Reviewer AVUJ (1/2)**
>
> We thank the reviewer for helpful feedback and address the comments below.
>
> ---
>
> > 1. [Significance] I think the paper makes the point for releasing more non-equilibrium structures (although, this is already the case in OC20 and OC22 if I understood well), which is important to state clearly to the community.
>
> Thanks for recognizing this! Indeed, single-point DFT calculations on non-equilibrium structures are cheaper to calculate than full DFT relaxations. Many publicly available datasets (e.g. PCQM4Mv2 [1]) only release equilibrium structures while those structures are obtained by relaxing intermediate non-equilibrium structures. DeNS training can further improve performance and will benefit from the release of more non-equilibrium structure datasets.
>
> Reference:
> [1] Nakata et al. PubChemQC Project: A Large-Scale First-Principles Electronic Structure Database for Data-Driven Chemistry. Journal of Chemical Information and Modeling 2017.
>
> ---
>
> > 2. [Weakness – Originality] The original idea is not groundbreaking: auxiliary tasks are known, the specific case of denoising as well, here the novelty is only in feeding also the forces as input.
>
> We compare the contributions of previous works and this work in **General Response 2**. In summary, the main contribution is that we generalize denoising to all atomistic datasets, including both non-equilibrium and equilibrium structures, with a simple modification of force encoding. All prior works on denoising do not encode forces, which is an ablation of our proposed approach and performs worse (Table 1(e) on OC20 and **General Response 4** on MD17).
>
> ---
>
> > 3. [Weakness – Quality] it is not always clear from the results shown in tables, that the proposed method improves the SOTA significantly.
>
> Training with DeNS improves state-of-the-art models on all the datasets we consider. The improvements are clear on MD17 (Table 5 and **General Response 3**), OC20 S2EF-2M (Table 1(d) and **General Response 5**) and OC20 S2EF-All+MD datasets (Table 2 and **General Response 5**). The performance gain is sometimes less on OC22 (Table 4 and **General Response 6**). The main point is that with this simple force encoding, we can generalize denoising to the broader set of non-equilibrium structures and improve all the state-of-the-art models.
>
> ---
>
> > 4. [Weakness – Quality] Since there are two measures of success (energy and forces), it's sometimes difficult to make a final decision.
>
> Please see **General Response 8**.
>
> ---
>
> > 5. [Weakness – Clarity] The paper has some typos.
>
> Thanks! We will fix them.
>
> ---
>
> > 6. [Weakness – Clarity] More space should be devoted to discussing results.
>
> We move the results of different hyper-parameters on OC22 (Table 4) to the appendix and incorporate some discussions on the results in **General Response 3, 5, 6** into the manuscript.
>
> ---
>
> > 7. [Weakness – Significance] Since the results are not strikingly better when using DeNS on the SOTA, and given it involves a number of additional hyperparameters (that obviously do not need very narrow fine-tuning, admittedly, but still, they involve more work and potential for problems), it is not clear yet whether the contribution would be used widely.
>
> We disagree on results not being substantially better than SoTA. To restate, please see **General Response 3** (MD17), **General Response 5** (OC20) and **General Response 6** (OC22).
>
> Any new method involves at least some additional hyper-parameters [1, 2, 3, 4]. The hyperparameters for DeNS are easily transferable across architectures without much tuning. Specifically, when training eSCN with DeNS onOC20 S2EF-2M (Table 1(d)), we use the same hyper-parameters as training EquiformerV2, and that achieves better performance, matching the results of EquiformerV2 without DeNS and saving 1.94$\times$ training time without any hyper-parameter tuning. Similarly, for MD17 in Table 5, we directly use the same hyper-parameters tuned for Equiformer and training with DeNS improves two architecture variants on all the molecules. Besides, some hyper-parameters such as $p_{\text{DeNS}}$ and $\sigma_{\text{high}}$ also apply across datasets like OC20 and OC22, and this makes tuning hyper-parameters easier.
>
> Reference:
> [1] Wang et al. Denoise pretraining on nonequilibrium molecules for accurate and transferable neural potentials. Journal of Chemical Theory and Computation, 2023.
> [2] Godwin et al. Simple GNN Regularisation for 3D Molecular Property Prediction and Beyond. ICLR 2022.
> [3] Zaidi et al. Pre-training via Denoising for Molecular Property Prediction. ICLR 2023.
> [4] Feng et al. Fractional Denoising for 3D Molecular Pre-training. ICML 2023.

---

> > ### Author Response · Authors · 2023-11-23
> > **Response to Reviewer AVUJ (2/2)**
> >
> > > 8. [Weakness – Significance] Table 1d is probably the most convincing result, to me (differences between models seem more significant). Is it computed on the validation set(s) ? (or train set ? I have a doubt).
> >
> > The force and energy MAEs in Table 1(d) are all on the validation set as mentioned in the caption.
> >
> > ---
> >
> > > 9. [Question] Trainings are very heavy.
> >
> > Our work improves sample efficiency, thus **reducing the training cost**. In Table 1(d) and Table 4, we show that training with DeNS is better than scaling up computationally expensive hyperparameters like $L_{max}$ and training epochs, resulting in 2$\times$ to 3$\times$ better training efficiency. The training time of baseline models has little to do with this work.
> >
> > ---
> >
> > > 10. [Question] There is some hyper-param tuning, and DeNS is used on top of the best models (EquiformerV2).
> >
> > Please refer to **7. in Response to Reviewer AVUJ** for hyper-parameter tuning.
> >
> > The reason we use the Equiformer series is because they are the current state-of-the-art and to see how well DeNS can improve training efficiency and performance compared to trivially scaling up computationally expensive hyper-parameters like maximum degrees and training epochs as done in their papers.
> >
> > ---
> >
> > > 11. [Question] Energy and force: sometimes only one is better, but as you say, it's a balance.
> >
> > Please see **General Response 8**.
> >
> >  ---
> >
> > > 12. [Question] Can you answer and/or improve the discussions (and/or presentation of results) in the Experiments section, to show that indeed, the improvement is significant? If some tables show no significant improvement, it should be explained why. If the method is mostly able to speed up training to achieve equivalent accuracy, state it (and it will weaken the paper's claim, but at the same time strengthen the submission).
> >
> > Please see **General Response 3** (MD17), **General Response 5** (OC20) and **General Response 6** (OC22). We will state the takeaways more clearly in the paper.
> >
> > ---
> >
> > > 13. [Question] Table 2 does not show a significant improvement from using DeNS.
> >
> > Please see **General Response 7**.
> >
> > ---
> >
> > > 14. [Ethics Concerns] Bad ecological impact because of many GPU-hours.
> >
> > The training time reported in this work is similar to that reported in previous works [1, 2]. The proposed method aims at and is capable of reducing the training time to achieve better results.
> >
> > Reference:
> > [1] Passaro et al. Reducing SO(3) Convolutions to SO(2) for Efficient Equivariant GNNs. ICML 2023.
> > [2] Liao et al. EquiformerV2: Improved Equivariant Transformer for Scaling to Higher-Degree Representations. ArXiv 2023.
> >
> > ---
> >
> > > 15. [Ethics Concerns] This research aims at shortening training time by using a smart data-augmentation strategy. But the results are very moderate, for an investment cost that is quite consequent.
> >
> > We disagree. DeNS improves performance on all datasets we considered. Please see **General Response 3** (MD17), **General Response 5** (OC20) and **General Response 6** (OC22) for discussions on the performance. The proposed method can improve both training efficiency and error instead of only aiming at shortening training time.
> >
> > Most of the investment in computational cost is one-time to exhaustively evaluate design choices for reporting in this paper. We believe the identified training recipe and hyperparameters are generally applicable and will help practitioners improve accuracy and reduce training cost on their datasets.

---

### Official Review · Reviewer_tmjX · 2023-11-01

**Soundness:** 2 fair
**Presentation:** 2 fair
**Contribution:** 3 good
**Rating:** 5
**Confidence:** 4

**Summary:**

In this paper, the authors proposed a denoising non-equilibrium structures (DeNS) training strategy to improve force field learning. Different from previous denoising approaches limited to equilibrium structures, DeNS enables the utilization of non-equilibrium structures for the denoising task by encoding the corresponding non-zero forces for specifying the denoising targets. Extensive experiments are conducted to demonstrate the effectiveness of DeNS.

**Strengths:**

- The target problem is of interest to the machine learning force field community.
- The proposed approach to encode the forces for specifying target structures when denoising perturbed non-equilibrium structures is a new modification compared to previous methods like noisy-node.
- The experimental evaluation covers both small and large-scale benchmarks. The ablation studies on the introduced hyperparameters are informative for practitioners to try the proposed DeNS approach in their applications.

**Weaknesses:**

- **Regarding the performance improvement** . Serving as a simple modification upon previous denoising training strategy on molecular modeling, performance improvement is the most important aspect to measure the quality of this work. However, there exist several issues that need to be further clarified:
  - The gains brought by DeNS diminish with the dataset scales up. From Table 1 and Table 2, we can see that models trained on the 2M subset of the OC20 S2EF dataset benefit a lot more from the DeNS auxiliary task compared to the OC20 S2EF-All+MD split.
  - The improvement on MD17 is limited compared to the OC series experiments.
  - The improvement on the IS2RE task is also limited (sometimes DeNS even hurts the performance).
  - The gains on the energy and force metrics are not consistent across different datasets with different scales. In the OC20 tasks, models trained with the DeNS task achieve lower force errors and competitive energy errors compared to the standard training and vice versa for the OC22 tasks.

The first issue relates to the scaling property of the proposed training strategy. I recommend the authors further design experiments to investigate whether such a phenomenon is due to the inability of the proposed DeNS to bring performance gains when large-scale data is provided or other factors of the model, optimization, and so on. The second issue relates to the generality of the proposed strategy. MD17 contains simple molecules instead of adsorbate-catalyst complex in OC20/22. This dataset is much smaller than OC20/22 dataset, on which the proposed DeNS is expected to bring more gains according to the phenomenon mentioned in the first issue. However, the results in Table 5 and 6 show the gains are limited. It is suggested to further verify the generality of DeNS. The third issue relates to the significance of the force field learning task. In OC and other similar applications, what we really care about is to obtain the property of the equilibrium states like relaxed energy and structure. The error of force field model is an indirect metric. In this sense, the significance of improvement brought by DeNS on energy and force error of S2EF task should be reexamined based on the IS2RE performance.

Overall, I recommend the authors to carefully clarify the proposed issues above with further experimental evidence to make some aspects of the proposed DeNS training strategy more clear for readers. I would like to increase my scores if the authors could address my concerns in this section and questions in the next section.

**Questions:**

1. In each iteration, the model uses either the standard training or the DeNS training. As DeNS training requires the forces to be encoded into the input atom features, I wonder how the force encoding module would be used in the standard training which instead uses the forces as labels.

2. In Table 5, the authors demonstrate that the DeNS training is more efficient and results in larger performance gains than increasing max degrees of irreps. I wonder why the authors changed the model from EquiformerV2 to EquiformerV1 for this investigation. After all, the EquiformerV2 model is claimed to largely benefit from scaling the max degrees of irreps. How would EquiformerV2 with different max degrees of irreps behave in the same setting of Table 5?

3. Could you provide more discussion on why you chose Equation 7 for the multi-scale noise scheduler?

---

> ### Author Response · Authors · 2023-11-23
> **Response to Reviewer tmjX (1/1)**
>
> We thank the reviewer for helpful feedback and address the comments below.
>
> ---
> > 1. [Weakness 1] The gains brought by DeNS diminish with the dataset scales up. From Table 1 and Table 2, we can see that models trained on the 2M subset of the OC20 S2EF dataset benefit a lot more from the DeNS auxiliary task compared to the OC20 S2EF-All+MD split.
>
> We update the results of training on the OC20 All+MD dataset (**General Response 7**) and discuss the performance gain on OC20 in **General Response 5**.
>
> ---
>
> > 2. [Weakness 2] The improvement on MD17 is limited compared to the OC series experiments.
>
> Please see **General Response 3**.
>
> ---
>
> > 3. [Weakness 3] The improvement on the IS2RE task is also limited (sometimes DeNS even hurts the performance).
>
> With the updated OC20 All+MD results, EquiformerV2 with DeNS is better on all validation and testing S2EF splits, and about the same on IS2RE, compared to EquiformerV2 without DeNS. Consistent with findings in prior works, where eSCN achieves lower energy and force MAE but higher IS2RE MAE than SCN, we find that improvements on S2EF don’t always translate to improvements on IS2RE. We posit that IS2RE performance might be bottlenecked by other design choices when running structural relaxations (optimizer settings, stopping criterion, etc.) and not S2EF performance. We will look into this in future work.
>
> For OC22, please see **General Response 6**.
>
> ---
>
> > 4.  [Weakness 4] The gains on the energy and force metrics are not consistent across different datasets with different scales. In the OC20 tasks, models trained with the DeNS task achieve lower force errors and competitive energy errors compared to the standard training and vice versa for the OC22 tasks.
>
> Please see **General Response 8**.
>
> ---
>
> > 5. [Question 1] In each iteration, the model uses either the standard training or the DeNS training. As DeNS training requires the forces to be encoded into the input atom features, I wonder how the force encoding module would be used in the standard training which instead uses the forces as labels.
>
> During standard training, we do not feed in the forces. Specifically, the force encoding module is an SO(3) linear layer, which is fed a tensor filled with all zeros during standard training. Force encoding is only used during DeNS training. We do not use any force labels in the validation and testing sets.
>
> ---
>
> > 6. [Question 2] In Table 5, the authors demonstrate that the DeNS training is more efficient and results in larger performance gains than increasing max degrees of irreps. I wonder why the authors changed the model from EquiformerV2 to EquiformerV1 for this investigation. After all, the EquiformerV2 model is claimed to largely benefit from scaling the max degrees of irreps. How would EquiformerV2 with different max degrees of irreps behave in the same setting of Table 5?
>
> We use Equiformer(V1) simply because EquiformerV2 [1] does not report results on MD17 and that makes the comparison difficult.
>
> The benefits of using higher degrees depend on datasets. As we can see in Table 5, on MD17, increasing $L_{max}$ from 2 to 3 in Equiformer(V1) leads to small gains in energy and force MAE. Therefore, this makes increasing $L_{max}$ further and using EquiformerV2 less well-motivated. It is likely that for this small dataset, using higher $L_{max}$ just leads to margin gains. However, training with DeNS achieves better energy and force errors across all the molecules and improves training efficiency by 3.1$\times$ compared to increasing $L_{max}$ from 2 to 3.
>
> [1] Liao et al. EquiformerV2: Improved Equivariant Transformer for Scaling to Higher-Degree Representations. ArXiv 2023.
>
> ---
>
> > 7. [Question 3] Could you provide more discussion on why you chose Equation 7 for the multi-scale noise scheduler?
>
> This was an empirical choice. We first experimented with a single noise scale (Index 4 in Table 1(e)) and then a multi-scale schedule similar to prior work in denoising score matching [1], which we found to work significantly better. We believe that a multi-scale schedule is more likely to span the distribution of meaningful non-equilibrium structures across a diverse range of atom types and geometries compared to a fixed $\sigma$. We add this intuition to the revision.
>
> [1] Song et al. Generative modeling by estimating gradients of the data distribution. NeurIPS 2019.

---

### Official Review · Reviewer_eCvt · 2023-11-02

**Soundness:** 3 good
**Presentation:** 3 good
**Contribution:** 2 fair
**Rating:** 5
**Confidence:** 4

**Summary:**

The paper introduces DeNS, an approach to improve energy and force predictions with the aid of non-equilibrium structures' denoising as an auxiliary task. Its implementation feeds forces from original structures as inputs, contributing to a well-structured problem. Demonstrated results indicate minor improvements on EquiformerV2 for datasets like OC20, OC22, and MD17.

**Strengths:**

-The paper addresses an essential challenge: the development of a self-supervised learning methodology using non-equilibrium molecules.

-The proposal offers a unique perspective on non-equilibrium denoising by discussing the ill-posed mapping. The handling of input encoding using forces seems logically feasible.

-The documentation is unambiguous and comprehensible.

**Weaknesses:**

-The paper could benefit from a broader theoretical discussion and a comparative analysis with other self-supervised techniques for non-equilibrium structures, such as denoising pretraining in [1], Noisy Nodes (using the OC20 dataset)[2], and improved noisy nodes (using MD17)[3]. These techniques have demonstrated efficacy for energy or force predictions for non-equilibrium molecules, hence their significance.

-The motivation behind the paper's approach needs additional validation.
a) The concept of “encoding force as input” finds extensive discussion in the paper. However, this approach needs corroborative proof from experiments. Results from Table 1e indicate energy prediction outcomes remain the same without force encoding. The possibility of label leakage and its contribution to improvement in force prediction, as discussed under question 2, needs examination.
b) Similar problem-solving approaches have been published. A comparative discussion highlighting the distinctions and superiority of this paper's proposed methodology would prove advantageous.

-The significant results were, to a large extent, achieved through Equiformer. Against Equiformer's backdrop, the improvements contributed by DeNS appear minimal.

[1] Yuyang Wang, Chang Xu, Zijie Li, and Amir Barati Farimani. Denoise pretraining on nonequilibrium molecules for accurate and transferable neural potentials. Journal of Chemical Theory and Computation, 2023.
[2] Feng, S., Ni, Y., Lan, Y., Ma, Z. &amp; Ma, W.. (2023). Fractional Denoising for 3D Molecular Pre-training. Proceedings of the 40th International Conference on Machine Learning, in Proceedings of Machine Learning Research 202:9938-9961 Available from https://proceedings.mlr.press/v202/feng23c.html.
[3] Jonathan Godwin, Michael Schaarschmidt, Alexander L Gaunt, Alvaro Sanchez-Gonzalez, Yulia Rubanova, Petar Velickovi ˇ c, James Kirkpatrick, and Peter Battaglia. Simple GNN regularisation for 3d molecular property prediction and beyond. In International Conference on Learning Representations, 2022.

**Questions:**

-While encoding input forces can mitigate the non-equilibrium denoising's ill-posed problem, [1] shows that denoising without force input is also plausible. Does this undermine your motivation and imply the redundancy of input force encoding?
-Could there be label leakage when encoding forces as input for energy and force predictions of structures?
-Given that force prediction is one of your experiments, should you consider adding the force prediction loss to eq. (6)?
-Provision of the pseudocode for DeNS training would be beneficial. This can offer insights into the usage of Multi-Scale Noise and other hyperparameters like p_{ DeNS }.

---

> ### Author Response · Authors · 2023-11-23
> **Response to Reviewer eCvt (1/2)**
>
> We thank the reviewer for helpful feedback and address the comments below.
>
> ---
> > 1. [Weakness 1] The paper could benefit from a broader theoretical discussion and a comparative analysis with other self-supervised techniques for non-equilibrium structures, such as denoising pretraining in [1], Noisy Nodes (using the OC20 dataset) [2], and improved noisy nodes (using MD17) [3]. These techniques have demonstrated efficacy for energy or force predictions for non-equilibrium molecules, hence their significance.
>
> Please see **General Response 2** for the comparison to previous works.
>
> ---
>
> > 2. Could there be label leakage when encoding forces as input for energy and force predictions of structures?
>
> No, there is no label leakage for forces since we do not encode any force labels during evaluation (i.e., on the validation and testing sets). We only use force labels during denoising on the training set. In fact, on the OC20 testing set, the energy and force labels are not even publicly available. Evaluation on the testing sets is performed by a centralized server on EvalAI maintained by the OC20 authors.
>
> To clarify the training procedure, we have both the upper blue block and the lower red block in Figure 2 during training but only have the upper blue block when evaluating the results on the validation and testing sets.
>
> We clarify this in the revision.
>
> ---
> > 3. [Weakness 2] The concept of “encoding force as input” needs corroborative proof from experiments. Results from Table 1e indicate energy prediction outcomes remain the same without force encoding.
>
> We conduct additional ablations with and without force encoding on MD17 in **General Response 4**. In summary, denoising with force encoding achieves the best results across all molecules. Denoising without force encoding is worse than no denoising for most molecules and is worse than with force encoding.
>
> On OC20 S2EF-2M (Table 1(e)), indeed the energy MAE is the same with and without force encoding, but the force MAE is significantly worse (~4%) without force encoding. We should look at both energy and force prediction at the same time when comparing two methods since there is always a trade-off between energy MAE and force MAE. For example, we can slightly increase energy coefficient so that we further reduce energy MAE but increase force MAE. In this case, we can have both lower energy and force MAE when we use force encoding. Moreover, using force encoding also leads to significantly better sample efficiency – training DeNS with force encoding for 12 epochs improves force MAE by 4% compared to without force encoding (Index 1 and Index 2 in Table 1(e)), as opposed to training EquiformerV2 without DeNS for 20 epochs instead of 12 (3% better force MAE for 66% more compute, Table 1(d)), suggesting that force encoding is quite important.
>
> ---
>
> > 4. [Weakness 2] Similar problem-solving approaches have been published. A comparative discussion highlighting the distinctions and superiority of this paper's proposed methodology would prove advantageous.
>
> Please see **General Response 2** for a comparison to previous works.
>
> The main difference in our denoising formulation lies in that we additionally encode forces as input. This generalizes denoising to non-equilibrium structures since the atomic forces help identify the unique non-equilibrium structure to denoise to. Without conditioning on forces, the problem of denoising non-equilibrium structures is ill-posed since there are many possible target structures to denoise to (see Figure 1).
>
> Previous works [1, 2, 3] are designed for equilibrium structures and are the same as our ablation of denoising without force encoding. While [4] applies the same approach to non-equilibrium structures in ANI-1 and ANI-1x datasets, we show that denoising non-equilibrium structures without force encoding is worse than with force encoding, and can sometimes even be worse than training without denoising altogether (**General Response 4**).
>
> Reference:
> [1] Godwin et al. Simple GNN Regularisation for 3D Molecular Property Prediction and Beyond. ICLR 2022.
> [2] Zaidi et al. Pre-training via Denoising for Molecular Property Prediction. ICLR 2023.
> [3] Feng et al. Fractional Denoising for 3D Molecular Pre-training. ICML 2023.
> [4] Wang et al. Denoise pretraining on nonequilibrium molecules for accurate and transferable neural potentials. Journal of Chemical Theory and Computation 2023.

---

> > ### Author Response · Authors · 2023-11-23
> > **Response to Reviewer eCvt (2/2)**
> >
> > > 5. [Weakness 3] The significant results were, to a large extent, achieved through Equiformer. Against Equiformer's backdrop, the improvements contributed by DeNS appear minimal.
> >
> > We respectfully disagree. Equiformer is the current state-of-the-art in equivariant Transformers for atomistic modeling, but is computationally expensive. The reason we first apply DeNS to Equiformer is to see how well DeNS can improve training efficiency and performance, compared to trivially scaling up computationally expensive hyper-parameters like maximum degrees and training epochs as was done in their paper.
> >
> > We show that DeNS improves sample efficiency and performance across datasets – MD17 (**General Response 3**), OC20 (**General Response 5**), OC22 (**General Response 6**) – and architectures – Equiformer (MD17), SEGNN (MD17), EquiformerV2 (OC20) and eSCN (OC20) – demonstrating its generality.
> >
> > ---
> >
> > > 6. [Question 1] While encoding input forces can mitigate the non-equilibrium denoising's ill-posed problem, [1] shows that denoising without force input is also plausible. Does this undermine your motivation and imply the redundancy of input force encoding?
> >
> > We believe it doesn’t. The formulation in [1] is the same as our “no force encoding” ablation, which we show is worse than DeNS with force encoding on OC20 (Table 1(e)) and MD17 (**General Response 4**). Conceptually, without force encoding, the target of denoising non-equilibrum structures can be arbitrary and  ill-posed (see Figure 1).
> >
> > On some datasets where the non-equilibrium structures are closely related to the ones at equilibrium, denoising without force encoding can still work, especially if the gain of mapping from noisy distributions to data distributions outweighs the issue of the targets being ill-posed. In fact, we have a similar observation in Table 1(e), where training DeNS without force encoding (Index 2) is better than training without any denoising as in Table 1(d). Nevertheless, DeNS with force encoding is a safer and more general choice, enabling further improvement (Index 1 in Table 1(e)).
> >
> > Reference:
> > [1] Wang et al. Denoise pretraining on nonequilibrium molecules for accurate and transferable neural potentials. Journal of Chemical Theory and Computation, 2023.
> >
> > ---
> >
> > > 7. [Question 3] Given that force prediction is one of your experiments, should you consider adding the force prediction loss to eq. (6)?
> >
> > No, Equation 6 corresponds to the objective during DeNS (lower red block in Figure 2). Force prediction loss is already covered in Equation 1 (upper blue block in Figure 2). As stated in “Auxiliary Task” in Section 3.2.3., “Specifically, given a batch of structures, for each structure, we decide whether we optimize the objective of DeNS (Equation 6) or the objective of the original task (Equation 1).” When training DeNS (Equation 6), since we already encode forces, predicting forces would be trivial and thus we do not add the force prediction loss to Equation 6.
> >
> > ---
> >
> > > 8. [Question 4] Provision of the pseudocode for DeNS training would be beneficial. This can offer insights into the usage of Multi-Scale Noise and other hyperparameters like p_{ DeNS }.
> >
> > Thanks for the suggestion! We add this to the appendix (Section E).

---

### Author Response · Authors · 2023-11-23
**General Response (1/4)**

# General Response 1: Revised manuscript

We updated the paper based on reviewers’ comments. We moved Table 3 from the main paper to the appendix, and therefore the numbers of tables are different from the previous draft of the paper. All the author responses use the new numbers in the revised paper.

---

# General Response 2: Comparison between previous works and this work

We summarize the comparison between previous works [3, 4, 2, 1] and this work in chronological order as below.

[3] first proposes the idea of adding noise to 3D coordinates and then using denoising as an auxiliary task. The auxiliary task is trained along with the original task without relying on another large dataset. Their approach requires known equilibrium structures and therefore is limited to QM9 and OC20 IS2RE and is not applied to force prediction. For QM9, all the structures are at equilibrium, and for OC20 IS2RE, the target of denoising is the equilibrium relaxed structure. Denoising without force encoding is well-defined on both QM9 and OC20 IS2RE. In contrast, this work proposes using force encoding to generalize their approach to force prediction and non-equilibrium structures, which have much larger datasets than equilibrium ones. Force encoding can achieve better results on OC20 S2EF dataset without any overhead (Index 1 and Index 2 in Table 1(e)) and is necessary on MD17 dataset (**General Response 3**).

[4] adopts the denoising approach proposed by [3] as a pre-training method and therefore requires another large dataset containing unlabelled equilibrium structures for pre-training. The main difference is that they use denoising as a pre-training objective to leverage additional unlabeled equilibrium data while [3] and this work use denoising along with the original task and do not use any unlabeled data.

[2] follows the same practice of pre-training via denoising proposed by [4] and proposes a different manner of adding noise. Specifically, they separate noise into dihedral angle noise and coordinate noise and only learn to predict coordinate noise. However, dihedral angle noise requires tools like RDKit to obtain rotatable bonds and cannot be applied to other datasets like OC20 and OC22.

Although [4] and [2] report results of force prediction on MD17 dataset, they first pre-train models on PCQM4Mv2 dataset and then fine-tune the pre-trained models on MD17 dataset. We note that their setting is different from ours since we do not use any dataset for pre-training. As for fine-tuning on MD17 dataset, [4] simply follows the same practice in standard training. [2] explores fine-tuning with Noisy Nodes objectives [3], but the performance gain is much smaller than ours. Concretely, in Table 5 in [2], the improvement in force prediction on Aspirin is about 2.6% while ours is 20.8%. Therefore, despite the fact that [4, 2] show improvement in energy and force prediction (only on MD17), the performance gain is because of pre-training via denoising on equilibrium structures from the PCQM4Mv2 dataset. In contrast, we achieve better results through denoising on the same non-equilibrium MD17 dataset. We additionally note that as discussed in Section 3.2.4, we can combine the two methods – we can first pre-train on PCQM4Mv2 [4, 2] and then fine-tune with DeNS. Besides, [3] does not report any result on the MD17 dataset.

[1] uses the same pre-training method as [4] but applies it to ANI-1 and ANI-1x datasets, which contain non-equilibrium structures. However, [1] does not encode forces, and denoising non-equilibrium structures without force encoding can sometimes lead to worse results compared to training without denoising. Although they show better results without force encoding, they only apply it to ANI-1 and ANI-1x datasets and it is unclear whether the results can be generalized to other datasets. Moreover, the results can be better with force encoding. [1] also report results on MD22, but they only compare the results with and without pre-training on ANI-1 and ANI-1x datasets.

In brief, our work explores how to denoise non-equilibrium structures with force encoding and therefore generalizes the method of [3] to a much larger set of non-equilibrium structures. Following [3], we use denoising as an auxiliary task instead of pre-training [4, 2, 1]. We show that denoising non-equilibrium structures with force encoding can achieve better results for all the cases we consider.

We include the discussion in the appendix of the revision (Section A.2).

Reference:
[1] Wang et al. Denoise pretraining on nonequilibrium molecules for accurate and transferable neural potentials. Journal of Chemical Theory and Computation 2023.
[2] Feng et al. Fractional Denoising for 3D Molecular Pre-training. ICML 2023.
[3] Godwin et al. Simple GNN Regularisation for 3D Molecular Property Prediction and Beyond. ICLR 2022.
[4] Zaidi et al. Pre-training via Denoising for Molecular Property Prediction. ICLR 2023.

---

### Author Response · Authors · 2023-11-23
**General Response (2/4)**

# General Response 3: Performance gain on MD17

First, in Table 4, comparing Equiformer ($L_{max} = 2$) and Equiformer ($L_{max} = 2$) + DeNS, we achieve overall better energy prediction and reduce force error by 12.8% while only increasing training time by 10.5%. This is 3.1$\times$ more efficient and has better results than increasing $L_{max}$ from 2 to 3.

Second, the gains from DeNS (as an auxiliary task) are comparable to pre-training as in [1]. For force prediction on MD17 dataset, [1] uses TorchMD-NET and pre-trains on the PCQM4Mv2 dataset, and they only report results on Aspirin and the improvement is about 17.2% (Table 3 in [1]). DeNS with EquiformerV1 has 20.8% improvement without relying on another dataset. Note that we only increase training time by 10% while their method needs to first pre-train on PCQM4Mv2 (**3000$\times$ larger than MD17**) and therefore takes much more time.

Overall, the performance gains of DeNS on MD17 are quite significant.

Reference:
[1] Zaidi et al. Pre-training via Denoising for Molecular Property Prediction. ICLR 2023.

---

# General Response 4: Denoising non-equilibrium structures with and without force encoding on MD17

We compare DeNS with and without force encoding on MD17 to demonstrate that force encoding is critical. The results are summarized below.
|     Molecule    |        | Equiformer ($L_{max}$ = 2) | + DeNS without force encoding | + DeNS with force encoding |
|:---------------:|--------|--------------------------|-------------------------------|----------------------------|
| Aspirin         | energy | 5.3                      | 8.6                           | **5.1**                    |
|                 | force  | 7.2                      | 9.1                           | **5.7**                    |
| Benzene         | energy | **2.2**                  | 2.3                           | 2.3                        |
|                 | force  | 6.6                      | 6.3                           | **6.1**                    |
| Ethanol         | energy | **2.2**                  | 2.3                           | **2.2**                    |
|                 | force  | 3.1                      | 3.3                           | **2.6**                    |
| Malondialdehyde | energy | 3.3                      | **3.2**                       | **3.2**                    |
|                 | force  | 5.8                      | 5.8                           | **4.4**                    |
| Naphthalene     | energy | **3.7**                  | 7.7                           | **3.7**                    |
|                 | force  | 2.1                      | 6.1                           | **1.7**                    |
| Salicylic Acid  | energy | 4.5                      | 5.2                           | **4.3**                    |
|                 | force  | 4.1                      | 10.6                          | **3.9**                    |
| Toluene         | energy | 3.8                      | 3.7                           | **3.5**                    |
|                 | force  | 2.1                      | 2.0                           | **1.9**                    |
| Uracil          | energy | 4.3                      | 5.5                           | **4.2**                    |
|                 | force  | **3.3**                  | 6.5                           | **3.3**                    |

DeNS with force encoding (rightmost column) achieves the best results across all molecules.

Compared to training without denoising, training with DeNS without force encoding only achieves slightly better results on some molecules (Benzene, Malondaldehyde, Toluene) and much worse results on others. For molecules on which training DeNS without force encoding is helpful, adding force encoding can further achieve even better results. For other molecules, force encoding is critical for DeNS to be effective.

In summary, training with DeNS with force encoding is necessary to achieve the best results, significantly outperforming training with denoising without force encoding or training without denoising altogether. We show this for MD17 in the table above, and for OC20 in Table 1(e) in the paper.

We add the results in the appendix of the revision (Section D.3).

---

### Author Response · Authors · 2023-11-23
**General Response (3/4)**

# General Response 5: Performance gain on OC20

For OC20 S2EF-2M dataset (Table 1(d)), we have shown that training EquiformerV2 with DeNS for 12 epochs is better than training EquiformerV2 without DeNS for 30 epochs and saves 2.33$\times$ training time. Moreover, eSCN trained with DeNS for 20 epochs can match the performance of EquiformerV2 trained without DeNS for 30 epochs and saves 1.91$\times$ training time. We note that when trained on OC20 S2EF-2M dataset, as shown in Figure 4 in [1], EquiformerV2 at most improves the training time by 2$\times$ compared to eSCN [2]. Therefore, the improvement is comparable. We have summarized these results below.

| Method              | Epochs | Force MAE | Energy MAE | Training time (GPU-hours) |
|---------------------|:------:|:---------:|:----------:|:-------------------------:|
| EquiformerV2        |   30   |   19.42   |     278    |            3495           |
| EquiformerV2 + DeNS |   12   |   19.32   |     271    |            1501           |
| eSCN + DeNS         |   20   |   19.07   |     279    |            1829           |

For OC20 S2EF-All+MD dataset, we updated the results as in **General Response 7**. EquiformerV2 trained with DeNS achieves better S2EF results and comparable IS2RE results. The improvement is not as significant as that on MD17 and OC20 S2EF-2M datasets since the OC20 S2EF-All+MD training set contain much more structures along relaxation trajectories, making new 3D geometries generated by DeNS less helpful. However, DeNS is valuable because most datasets are not as large as OC20 S2EF-All+MD dataset but have sizes closer to MD17 and OC20 S2EF-2M datasets.

Reference:
[1] Liao et al. EquiformerV2: Improved Equivariant Transformer for Scaling to Higher-Degree Representations. ArXiv 2023.
[2] Passaro et al. Reducing SO(3) Convolutions to SO(2) for Efficient Equivariant GNNs. ICML 2023.

---

# General Response 6: Performance gain on OC22

On OC22 S2EF-Total, training with DeNS improves all metrics except the OOD force MAE on the validation set. Moreover, for IS2RE-Total, the improvement from training with DeNS on only OC22 is comparable to that of training on much larger OC20 + OC22 datasets in previous works. We have summarized these results in the table below. Specifically, in the top two rows in Table 3, training GemNet-OC on both OC20 and OC22 datasets (about 138M + 8.4M structures) improves IS2RE-Total energy MAE ID by 129meV and OOD by 50meV compared to training GemNet-OC on only OC22 dataset (8.4M structures). In the last two rows, compared to training without DeNS, training EquiformerV2 with DeNS improves ID by 90meV and OOD by 48meV. Thus, training with DeNS significantly improves sample efficiency and performance on OC22.

| Method              | Training set | IS2RE-Total MAE (ID) | IS2RE-Total MAE (OOD) |
|---------------------|:------------:|:--------------------:|:---------------------:|
| GemNet-OC           |     OC22     |         1329         |          1584         |
| GemNet-OC           |  OC20 + OC22 |         1200         |          1534         |
| EquiformerV2        |     OC22     |         1119         |          1440         |
| EquiformerV2 + DeNS |     OC22     |         1029         |          1392         |

---

### Author Response · Authors · 2023-11-23
**General Response (4/4)**

# General Response 7: Update results on OC20 S2EF-All+MD dataset

We updated the results on OC20 S2EF-All+MD dataset in Table 2. EquiformerV2 trained with DeNS achieves better performance on all the metrics and sets new state-of-the-art results.

|                                     | S2EF validation |           |  S2EF test |           | IS2RE test |
|-------------------------------------|:---------------:|:---------:|:----------:|:---------:|:----------:|
| Method                              |    Energy MAE   | Force MAE | Energy MAE | Force MAE | Energy MAE |
| EquiformerV2 ($\lambda_E = 4$)        |       227       |    15.0   |     219    |    14.2   |     309    |
| EquiformerV2 + DeNS ($\lambda_E = 4$) |     **221**     |  **14.2** |   **216**  |  **13.4** |   **308**  |

---

# General Response 8: Improvement on energy and force errors. Sometimes only one is better. The gains on the energy and force metrics are not consistent.

We generally agree that the gains on each metric depends on the dataset. However, the improvements with DeNS are consistent with those from prior advances in architectures. Specifically, for OC20, better architectures typically improve force error more than energy error, which is the same case as DeNS (Table 1(d)). For OC22, on the other hand, the improvement brought by better architectures on energy error is larger than that on force error, which is consistent with the performance gain of DeNS (Table 3). We do not claim that improvements across energy and force errors should be consistent. Also, there is often a trade-off in energy and force errors, modulated by the respective loss coefficients. Thus, slight adjustments in these coefficients can achieve lower errors for both metrics.

Additionally, being comparable on one metric (i.e., energy or force) and better on the other (i.e., force or energy) still implies an overall improvement over the baselines.

---

### Author Response · Authors · 2023-11-23
**Code for anonymous review**

As mentioned in the section of Reproducibility Statement, we provide the link to an anonymous repository here: https://anonymous.4open.science/r/dens-iclr2024-E08F/README.md

---

### Meta-Review · Area_Chair_4kyp · 2023-12-05

**Metareview:**

In this work, the authors introduced a novel training strategy called denoising non-equilibrium structures (DeNS) to enhance the learning of force fields. Unlike conventional denoising methods focused on equilibrium structures, DeNS allows the incorporation of non-equilibrium structures in the denoising process. This is achieved by encoding the associated non-zero forces to define the denoising targets.

The reviewers raised multiple concerns regarding the paper, including the novelty and motivation(reviewer avuj and ecvt) and the empirical benefit(all the reviewers during the discussion period). All these points are critical to the paper's judgment. Given the unaddressed issues, I recommend rejection and suggest the authors submit the work to a future venue.

**Justification For Why Not Higher Score:**

See the metareview

**Justification For Why Not Lower Score:**

N/A

---

### Decision · Program_Chairs · 2024-01-16

Reject